# Dynamic interactions between the RNA chaperone Hfq, small regulatory RNAs, and mRNAs in live bacterial cells

**Seongjin Park[1‡], Karine Prévost[2‡], Emily M Heideman[1†], Marie-Claude Carrier[2†], Muhammad S Azam[1], Matthew A Reyer[3], Wei Liu[1], Eric Massé[2]\*, Jingyi Fei[1,3]\***

[1]Department of Biochemistry and Molecular Biology, The University of Chicago, Chicago, United States; [2]RNA Group, Department of Biochemistry, University of Sherbrooke, Sherbrooke, Canada; [3]Institute for Biophysical Dynamics, The University of Chicago, Chicago, United States

**Abstract** RNA-binding proteins play myriad roles in regulating RNAs and RNA-mediated functions. In bacteria, the RNA chaperone Hfq is an important post-transcriptional gene regulator. Using live-cell super-resolution imaging, we can distinguish Hfq binding to different sizes of cellular RNAs. We demonstrate that under normal growth conditions, Hfq exhibits widespread mRNA-binding activity, with the distal face of Hfq contributing mostly to the mRNA binding in vivo. In addition, sRNAs can either co-occupy Hfq with the mRNA as a ternary complex, or displace the mRNA from Hfq in a binding face-dependent manner, suggesting mechanisms through which sRNAs rapidly access Hfq to induce sRNA-mediated gene regulation. Finally, our data suggest that binding of Hfq to certain mRNAs through its distal face can recruit RNase E to promote turnover of these mRNAs in a sRNA-independent manner, and such regulatory function of Hfq can be decoyed by sRNA competitors that bind strongly at the distal face.

**\*For correspondence:**
eric.masse@usherbrooke.ca (EM);
jingyifei@uchicago.edu (JF)

[†]These authors also contributed equally to this work
[‡]These authors also contributed equally to this work

**Competing interests:** The authors declare that no competing interests exist.

## Introduction

In all kingdoms of life, RNA-binding proteins (RBPs) orchestrate the post-transcriptional fates of RNAs by modulating their turnover, structure, and localization, and often as a companion of regulatory RNAs. As one of the most abundant RBPs in bacterial cells, Hfq is an important and prevalent post-transcriptional gene regulator (*Brennan and Link, 2007*; *Vogel and Luisi, 2011*; *Updegrove et al., 2016*). Acting as a chaperone of sRNA-mediated gene regulation, Hfq protects small RNAs (sRNAs) from degradation and promotes sRNA–mRNA duplex formation (*Brennan and Link, 2007*; *Vogel and Luisi, 2011*; *Updegrove et al., 2016*). Binding of sRNAs to target mRNAs further leads to changes in the translation activity and the stability of the mRNAs (*Storz et al., 2011*; *Wagner and Romby, 2015*). Moreover, other functions of Hfq in regulating translation and degradation of mRNAs independent of the sRNA-mediated regulatory pathway have also been reported (*Urban and Vogel, 2008*; *Vytvytska et al., 2000*; *Pei et al., 2019*; *Hajnsdorf and Régnier, 2000*; *Mohanty et al., 2004*). Loss of Hfq compromises the fitness of bacterial cells, especially under harsh conditions, and abolishes the virulence of bacterial pathogens (*Sobrero and Valverde, 2012*; *Tsui et al., 1994*).

Hfq binds broadly to cellular mRNAs, sRNAs, and ribosomal RNAs (rRNAs) (*Sittka et al., 2008*; *Tree et al., 2014*; *Chao et al., 2012*; *Melamed et al., 2016*; *Melamed et al., 2020*; *Andrade et al., 2018*; *Dos Santos et al., 2020*). Hfq interacts with RNAs through multiple interfaces of its homo-hexameric structure. The surface containing the N-terminal α-helices is referred to as the 'proximal face', whereas the opposite surface is referred to as the 'distal face', and the outer ring as the 'rim' (*Figure 1a*). The proximal face preferably binds U-rich sequences, while the distal face prefers A-rich

**eLife digest** Messenger RNAs or mRNAs are molecules that the cell uses to transfer the information stored in the cell's DNA so it can be used to make proteins. Bacteria can regulate their levels of mRNA molecules, and they can therefore control how many proteins are being made, by producing a different type of RNA called small regulatory RNAs or sRNAs. Each sRNA can bind to several specific mRNA targets, and lead to their degradation by an enzyme called RNase E. Certain bacterial RNA-binding proteins, such as Hfq, protect sRNAs from being degraded, and help them find their mRNA targets.

Hfq is abundant in bacteria. It is critical for bacterial growth under harsh conditions and it is involved in the process through which pathogenic bacteria infect cells. However, it is outnumbered by the many different RNA molecules in the cell, which compete for binding to the protein. It is not clear how Hfq prioritizes the different RNAs, or how binding to Hfq alters RNA regulation. Park, Prévost et al. imaged live bacterial cells to see how Hfq binds to RNA strands of different sizes.

The experiments revealed that, when bacteria are growing normally, Hfq is mainly bound to mRNA molecules, and it can recruit RNase E to speed up mRNA degradation without the need for sRNAs. Park, Prévost et al. also showed that sRNAs could bind to Hfq by either replacing the bound mRNA or co-binding alongside it. The sRNA molecules that strongly bind Hfq can compete against mRNA for binding, and thus slow down the degradation of certain mRNAs.

Hfq could be a potential drug target for treating bacterial infections. Understanding how it interacts with other molecules in bacteria could provide help in the development of new therapeutics. These findings suggest that a designed RNA that binds strongly to Hfq could disrupt its regulatory roles in bacteria, killing them. This could be a feasible drug design opportunity to counter the emergence of antibiotic-resistant bacteria.

sequences, with the exact composition of the A-rich motif varying from species to species (*Horstmann et al., 2012*; *Link et al., 2009*; *Robinson et al., 2014*; *Salim et al., 2012*; *Someya et al., 2012*). The rim can also interact with UA-rich RNAs through the patch of positively charged residues (*Dimastrogiovanni et al., 2014*; *Murina et al., 2013*; *Peng et al., 2014*). Finally, the unstructured C-terminal end of Hfq can also interact with certain RNAs to promote the exchange of RNAs (*Robinson et al., 2014*; *Dimastrogiovanni et al., 2014*; *Santiago-Frangos et al., 2016*). The most refined model describing the interactions between Hfq and sRNAs/mRNAs has sorted sRNAs into two classes (*Schu et al., 2015*). The proximal face of Hfq is generally important for the binding of the sRNAs through their poly-U tail of the Rho-independent terminator, independent of sRNA class. Class I sRNAs (such as RyhB and DsrA) then use the rim as the second binding site, whereas class II sRNAs (such as ChiX and MgrR) use the distal face as the second binding site (*Schu et al., 2015*). In addition, the preferred target mRNAs of the two classes of the sRNA are proposed to have the complementary binding sites on Hfq, *i.e.*, class I sRNA-targeted mRNAs (or class I mRNAs) binding to the distal face, and class II sRNA-targeted mRNAs (or class II mRNAs) binding to the rim, to efficiently form sRNA–mRNA complexes (*Schu et al., 2015*). As a pleiotropic regulator, Hfq establishes additional interactions with other essential protein factors. Particularly, RNase E, the key ribonuclease in the bacterial inner membrane for processing and turnover of rRNAs and mRNAs, is known to interact with Hfq through its C-terminal scaffold region (*Bruce et al., 2018*; *Ikeda et al., 2011*; *Morita et al., 2005*; *Worrall et al., 2008*). The Hfq-RNase E interactions can promote the degradation of the sRNA-targeted mRNA (*Morita et al., 2005*; *Afonyushkin et al., 2005*; *Pfeiffer et al., 2009*; *Prévost et al., 2011*).

While Hfq is an abundant RBP in bacterial cells (*Ali Azam et al., 1999*; *Kajitani et al., 1994*), it is still considered to be limiting, given the abundance of cellular RNAs. Particularly, in vitro studies on specific sRNAs demonstrate that Hfq binds RNAs tightly with a dissociation constant in the range of nanomolar, and the Hfq–RNA complexes are stable with a lifetime of >100 min (*Fender et al., 2010*; *Salim and Feig, 2010*; *Olejniczak, 2011*). However, under stress conditions, induced sRNAs can regulate target mRNAs within minutes, raising a long-standing question of how sRNAs can rapidly access Hfq that might be tightly bound by pre-existing cellular RNAs. To address this question, a model of RNA exchange on Hfq, that is, an RNA can actively displace another RNA from Hfq, was

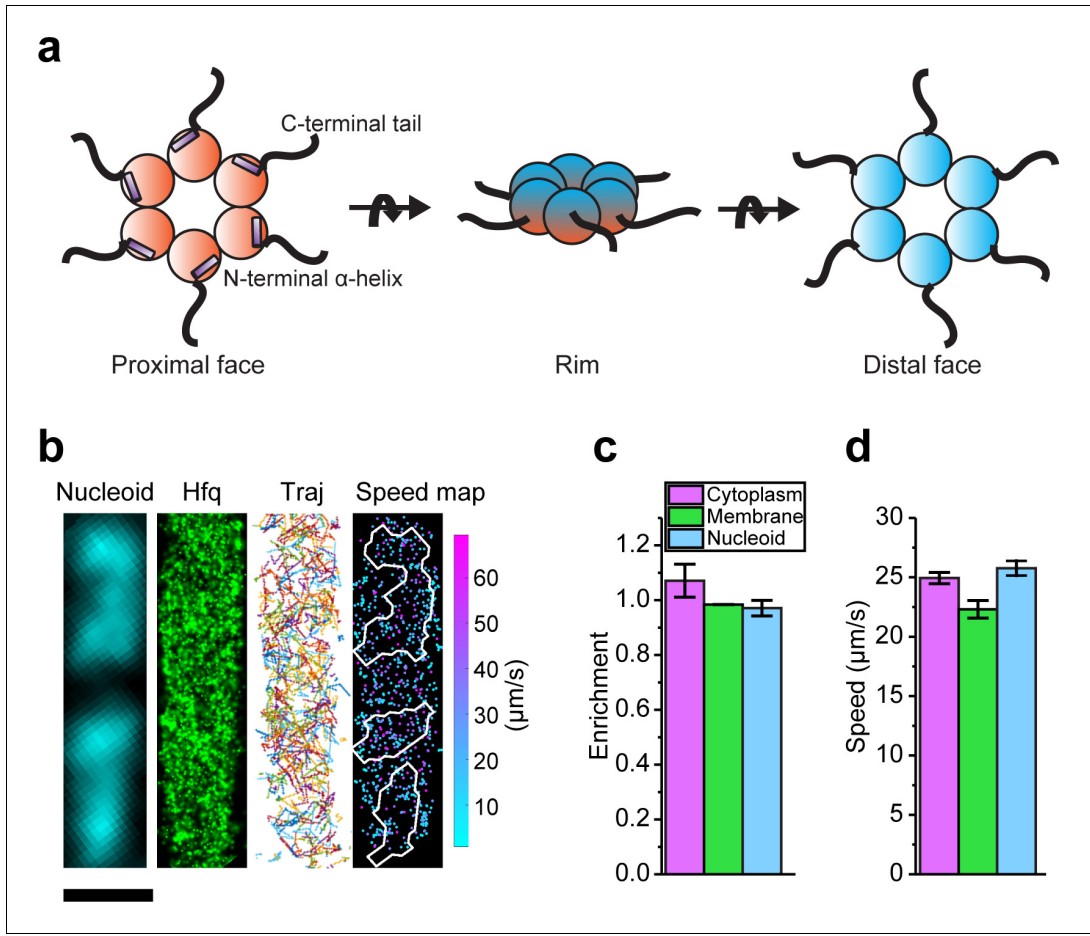

**Figure 1.** Diffusion and localization of Hfq during exponential growth. (**a**) Schematic representation of Hfq with three RNA binding faces indicated. (**b**) A representative example of WT Hfq-mMaple3 in WT *rne* background in a single cell during exponential growth under no treatment (NT) condition. Nucleoid is stained with Hoechst in live cells. 2D reconstructed image of Hfq-mMaple3 is shown in the black background. Different diffusion trajectories from tracking algorithm are shown in different colors ('Traj'). One-step displacement (*osd*) speed map (unit: µm/s) is shown as a scatter plot where different colors represent different speeds at each position, and the white curves represent the nucleoid regions detected by Hoechst staining. The scale bar represents 1 µm. (**c**) Enrichment of Hfq localization is calculated for cytoplasm, membrane, and nucleoid regions under NT condition. (**d**) Average *osd* speed of Hfq within the cytoplasm, membrane, and nucleoid regions under NT condition. Error bars in all plots represent the standard deviation (s.d.) from two experimental replicates, with each data set containing ~20,000 trajectories from ~80 cells.

The online version of this article includes the following source data and figure supplement(s) for figure 1:

**Source data 1.** Single cell speed (**b**) and average enrichment and *osd* speed (**c** and **d**).
**Figure supplement 1.** mMaple3 tag on Hfq does not affect growth rate.
**Figure supplement 1—source data 1.** Growth curves.
**Figure supplement 2.** mMaple3 tag on Hfq does not affect mRNA degradation by sRNA.
**Figure supplement 2—source data 1.** Densitometry analysis of northern blots (**c**).
**Figure supplement 3.** Fixed cells as the stationary control for tracking analysis.
**Figure supplement 3—source data 1.** *Osd* speed distribution (**b**), MSD plots (**c**), and *osd* speed of a single cell (**d**).

proposed to account for the fast sRNA-mediated stress response (*Vogel and Luisi, 2011*; *Updegrove et al., 2016*; *Wagner, 2013*). While in vitro biophysical experiments can measure the affinity of RBP binding to many different RNAs under many different controllable conditions, it is difficult to replicate the concentrations, compartmentalization, crowding, competitive binding, and changes in cellular conditions that can affect the behavior and function of RBPs in live cells.

Therefore, the mechanism(s) that can recycle Hfq from pre-bound RNAs in live cells remains to be elucidated.

To address this question in a cellular context, we measured the diffusivity of Hfq in live *Escherichia coli* cells, using single-molecule localization microscopy (SMLM) (*Manley et al., 2008*), with a rationale that the diffusivity is affected by the molecular weight of the molecules (*Mika and Poolman, 2011*), and therefore can report the interactions between Hfq with different cellular components. By measuring Hfq diffusivity under a variety of cellular conditions in combination with other biochemical assays, we demonstrate that Hfq dynamically changes its interactions with different RNAs. Specifically, the two classes of sRNA can gain access to mRNA pre-bound Hfq through different mechanisms. Finally, our data suggest that binding of Hfq to certain mRNAs through its distal face can recruit RNase E to promote turnover of these mRNAs in a sRNA-independent manner.

## Results

### Cellular Hfq freely diffuses in the absence of stress

Hfq was tagged by a photo-switchable fluorescent protein (FP), mMaple3 (*Wang et al., 2014*), at the C-terminus, and the fused *hfq* gene was integrated into the genomic locus to replace the wild-type (WT) *hfq* (denoted as '*hfq-mMaple3*', Materials and methods). The strain harboring *hfq-mMaple3* showed comparable growth curve as the WT strain, whereas Δ*hfq* strain showed a growth defect (*Figure 1—figure supplement 1*). In addition, Hfq-mMaple3 showed activity comparable to WT Hfq protein, as revealed by northern blots of RyhB-mediated *sodB* mRNA degradation and MicA-mediated *ompA* mRNA degradation (*Figure 1—figure supplement 2*).

We performed single-particle tracking using SMLM in two dimensions (2D). Images were collected at a rate of 174 frames per second with 2.4 ms exposure time for each frame. Imaging conditions and parameters for applying tracking algorithm were optimized using fixed samples as the control (*Figure 1—figure supplement 3*). We first tracked Hfq-mMaple3 in live cells grown at exponential phase (referred to as '<u>n</u>o <u>t</u>reatment', or 'NT' case). In the NT condition, Hfq-mMaple3 exhibited a relatively uniform distribution within the cell (*Figure 1b*), consistent with the distribution revealed by the earlier live-cell imaging with Hfq tagged by a different FP (Dendra2) (*Persson et al., 2013*). Quantification of Hfq-mMaple3 localization with DNA stained by Hoechst revealed a slightly higher cytoplasm enrichment than nucleoid or membrane localization in the NT condition (*Figure 1c*). We did not observe a helical organization along the longitudinal direction of the bacterial cell (*Taghbalout et al., 2014*), membrane localization (*Diestra et al., 2009*), or cell pole localization (*Kannaiah et al., 2019*), as reported in a few fixed-cell experiments. In addition, we calculated the one-step displacement (*osd*) speed of individual Hfq-mMaple3 protein at each time step and plotted *osd* speed as a function of the cellular coordinate in a diffusivity speed map (*Figure 1b*). The speed map and quantification of the average *osd* speed suggest that Hfq diffuses similarly within the nucleoid and cytoplasmic region, but slightly more slowly in the membrane (*Figure 1d*), which could be due to the association with RNase E in the inner membrane.

### Binding of mRNAs to Hfq decreases its diffusivity primarily through the distal face of Hfq

We first tested the effect of mRNA on Hfq-mMaple3 diffusivity by treating the cells with rifampicin (*Figure 2a*). Rifampicin is an antibiotic that inhibits transcription and results in the loss of most cellular RNAs. We estimated the effective diffusion coefficient (D) by fitting the linear function to the mean squared displacement (MSD) as a function of time lag (Δt) (*Figure 2b*). Transcription inhibition increased the diffusivity of Hfq-mMaple3 (*Figure 2b*), suggesting that a fraction of Hfq-mMaple3 proteins is associated with cellular RNAs, consistent with a previous report (*Persson et al., 2013*). We also generated several control constructs: mMaple3 protein alone, mMaple3 fused to an engineered synthetic antibody sAB-70 (*Mukherjee et al., 2018*), and mMaple3-fused scFv-GCN4 used in the SunTag imaging approach (*Tanenbaum et al., 2014*), which are not reported to bind to any RNA. None of them showed any changes in diffusivity upon rifampicin treatment (*Figure 2—figure supplement 1*), confirming that the change in diffusivity is not due to the change in the cellular milieu. It should be noted that under the current rifampicin treatment (200 µg/mL concentration for 15 min), mRNAs, which have an average half-live of 1–4 min (*Chen et al., 2015*), are preferentially

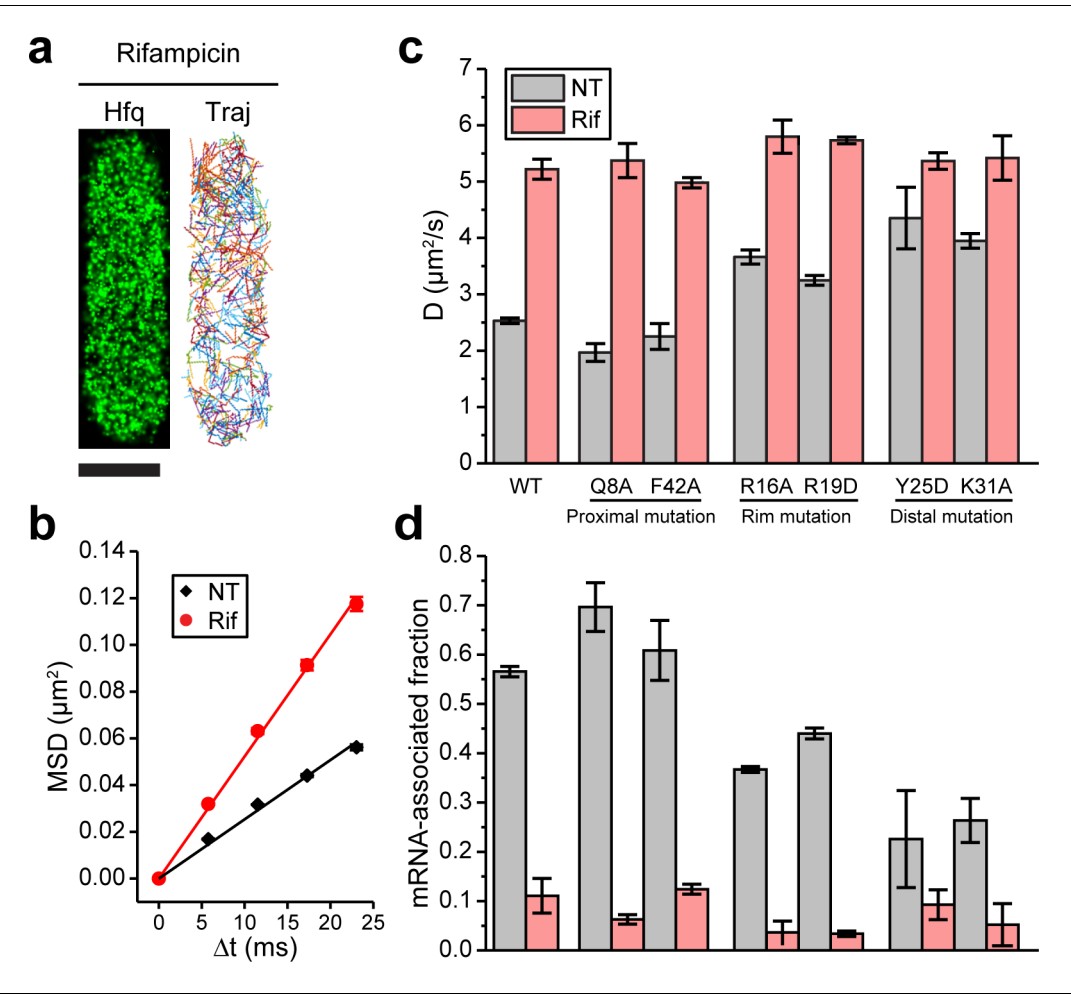

**Figure 2.** Binding of mRNAs to Hfq decreases its diffusivity primarily through the distal face of Hfq. (a) A representative example of Hfq-mMaple3 with rifampicin treatment (Rif) in a single cell. 2D reconstructed image is shown in the black background (left), and different diffusion trajectories are shown in different colors (right). The scale bar represents 1 µm. (b) Mean squared displacement (MSD) is plotted against the time lag ($\Delta t$) for Hfq-mMaple3 under NT and Rif cases. The linear fitting lines are shown. (c) Ensemble diffusion coefficients are plotted for WT and six mutants of Hfq-mMaple3 under NT and rifampicin treatment conditions. (d) mRNA-associated fraction for WT and six mutants of Hfq under NT and Rif conditions. Error bars in all plots represent the s.d. from two or three experimental replicates, with each data set containing ~5,000 trajectories (for D value calculation) or ~20,000 trajectories (for mRNA-associated fraction calculation) from ~100 cells. All fitting results are reported in *Supplementary file 1*.

The online version of this article includes the following source data and figure supplement(s) for figure 2:

**Source data 1.** Mean squared displacement (MSD) plot for NT and Rif (b).
**Figure supplement 1.** Diffusivities of different mMaple3 controls are not affected by treatment with rifampicin.
**Figure supplement 1—source data 1.** Mean squared displacement (MSD) plots for mMaple3 controls (b).
**Figure supplement 2.** Estimation of mRNA-associated Hfq fraction.
**Figure supplement 2—source data 1.** CDFs, PDFs, and fits of *osd²* of Hfq (a, b, and c) and comparison of *osd* D and mean squared displacement (MSD) D (d).

degraded, compared to tRNAs (*Svenningsen et al., 2017*) and rRNAs (*Blundell and Wild, 1971*). While many sRNAs show long half-lives when target-coupled degradation is reduced in the absence of mRNAs upon rifampicin treatment (*Massé et al., 2003*; *Zhang et al., 2002*), some sRNAs do have short half-lives (*Vogel et al., 2003*). Therefore, rifampicin treatment might also reduce the fraction of Hfq bound by sRNAs. However, our data suggest that binding of sRNA to RNA-free Hfq or to mRNA-associated Hfq does not change the diffusion coefficients of corresponding species (see

sections below). Therefore, we interpreted that the change in the diffusion coefficient upon rifampicin treatment primarily reflected the binding of mRNAs to Hfq.

We next introduced point mutations on Hfq-mMaple3 that are shown to affect RNA binding: Q8A and F42A at the proximal face, R16A and R19D at the rim, and Y25D and K31A at the distal face (*Schu et al., 2015*; *Zhang et al., 2013*), and imaged these mutant Hfq-mMaple3 proteins under NT and rifampicin treated conditions. With rifampicin treatment, all Hfq mutants exhibited similar diffusivity. However, mutations on different faces changed Hfq diffusivity in the NT case to different levels, suggesting that mutations on different faces changed the ability of Hfq to bind cellular mRNAs. Specifically, both proximal face mutants (Q8A and F42A) exhibited similar diffusivity as the WT Hfq-mMaple3; both rim mutations (R16A and R19D) had a minor increase in diffusivity under NT condition; and both distal face mutations (Y25D and K31A) led to the largest increase in the diffusivity under NT condition, with the diffusion coefficients close to the rifampicin treated case (*Figure 2c*). Comparison of the WT and the mutant Hfq-mMaple3 proteins supports conclusions that Hfq binds mRNAs in the cell and that binding of mRNAs is primarily achieved through the interactions with the distal face of Hfq, whereas the rim also contributes to the mRNA binding in a minor way.

## Majority of Hfq proteins are occupied by mRNAs in the cell during exponential growth

To analyze the subpopulations of Hfq-mMaple3 under different conditions, we plotted one-step squared displacement ($osd^2$) in a histogram. Consistent with D values, distribution of $osd^2$ overall shifted to larger values with rifampicin treatment compared to the NT case (*Figure 2—figure supplement 2*). We fit the cumulative probability density function (CDF) of $osd^2$ with double populations (*Figure 2—figure supplement 2*; *Bettridge et al., 2021*). WT Hfq and all mutants with rifampicin treatment showed consistently 86–98% fast population with average $osd^2$ of 0.048–0.053 $\mu m^2$, and the remaining 2–14% slow population with average $osd^2$ of 0.0022–0.0054 $\mu m^2$ (*Supplementary file 1*). In the NT case, WT Hfq showed (57 ± 1)% slow population and (43 ± 1)% fast population. Consistent with the previous interpretation (*Persson et al., 2013*), we assigned the slow population as the mRNA-associated fraction, and the fast population as mRNA-free fraction. This result is consistent with the previous hypothesis that Hfq proteins are largely occupied in the cell (*Vogel and Luisi, 2011*; *Updegrove et al., 2016*; *Wagner, 2013*). Y25D and K31A mutants had the most compromised mRNA binding ability, with (23 ± 10)% and (26 ± 4)% of the population being mRNA-associated under NT condition, respectively (*Figure 2d*). It is worth noting that as sRNA binding to RNA-free Hfq or to mRNA-associated Hfq does not change the diffusion coefficients of corresponding species (see sections below), it is possible that a subpopulation of mRNA-free or mRNA-associated Hfq might be sRNA-associated Hfq, or a sRNA-mRNA-Hfq ternary complex, respectively. In addition, the remaining 2–14% slow population after rifampicin treatment may reflect Hfq interactions with rRNAs or possibly DNA (*Malabirade et al., 2018*; *Orans et al., 2020*).

## sRNAs can access mRNA-associated Hfq in a face-dependent manner

We next examined the effect of sRNAs on the diffusivity of Hfq-mMaple3. We ectopically induced expression of different sRNAs, including RyhB, a class I sRNA, ChiX, a class II sRNA, and an sRNA that is less clearly defined between these two classes, SgrS (*Schu et al., 2015*) from the same vector. Whereas overexpression of RyhB or SgrS did not cause any noticeable changes in the Hfq-mMaple3 diffusivity or mRNA-associated fraction, overexpression of ChiX dramatically increased its diffusivity and lowered the mRNA-associated fraction (*Figure 3a and b*).

As described above, the distal face is the primary binding site for mRNAs in the cell (*Figure 2c and d*). Since ChiX requires binding at both the proximal and distal faces, we expected the diffusivity of Hfq to increase after shifting from mRNA-associated Hfq to ChiX-associated Hfq. Due to the relatively small molecular weight of sRNAs (~50–300 nucleotides in length), sRNA-associated Hfq-mMaple3 has similar diffusivity as free Hfq-mMaple3. We then checked if ChiX could compete with mRNAs for binding to Hfq in vitro using electrophoretic mobility shift assay (EMSA). A radiolabeled fragment of *ptsG* mRNA was pre-incubated with purified Hfq protein and then chased with unlabeled ChiX. Consistent with the in vivo results, ChiX can effectively displace *ptsG* from Hfq (*Figure 3a, b, and e*).

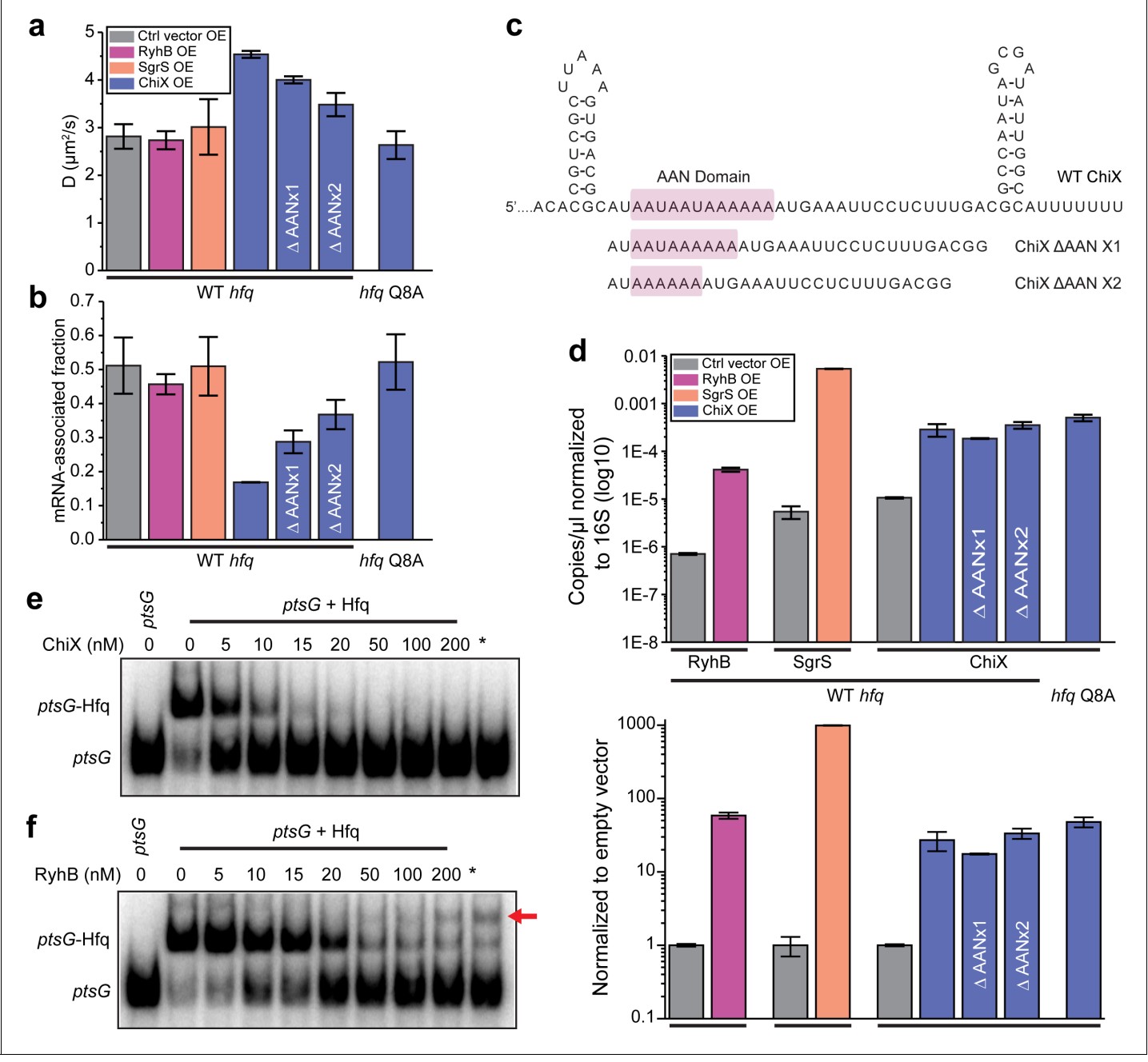

**Figure 3.** sRNAs can displace mRNA from Hfq in a face-dependent way. (**a**) Diffusion coefficients of WT Hfq-mMaple3 with control vector, RyhB, SgrS, WT ChiX, two ChiX mutants (with one or two AAN motif deleted) overexpressed, or Hfq Q8A-mMaple3 with WT ChiX overexpressed. All sRNAs were produced from an IPTG inducible promoter. (**b**) mRNA-associated fraction of Hfq, for the same cases as in (**a**). Error bars in all plots represent the s.d. from two experimental replicates, with each data set containing ~5,000 trajectories (for D value calculation) or ~20,000 trajectories (for mRNA-associated fraction calculation) from ~100 cells. (**c**) Sequences of WT ChiX and two ChiX mutants (with one or two AAN motif deleted). (**d**) ddPCR measurement of the abundance of RyhB, SgrS, WT ChiX, and ChiX ΔAANx1-2 mutants in the WT *hfq-mMaple3* and WT ChiX in the *hfq* Q8A-*mMaple3* background. The abundance of each sRNA is either normalized to the 16S rRNA (top) or to the control vector (bottom). Competition of (**e**) ChiX and (**f**) RyhB for mRNA-associated Hfq. 20 nM of a *ptsG* RNA fragment was pre-incubated with 100 nM Hfq before addition of increasing concentration of ChiX or RyhB sRNA. * marks the cases in which Hfq (100 nM) and ChiX or RyhB (100 nM) were simultaneously added to 20 nM *ptsG* fragment. Data is representative of three independent experiments. All fitting results are reported in *Supplementary file 1*.

The online version of this article includes the following source data and figure supplement(s) for figure 3:

**Source data 1.** ddPCR plots (**d**).

**Figure supplement 1.** Determination of $K_d$ values for Hfq interaction with RNAs.

*Figure 3 continued on next page*

*Figure 3 continued*

**Figure supplement 1—source data 1.** Quantification of electrophoretic mobility shift assay (EMSA) results.
**Figure supplement 2.** Quantification of sRNA expression by FISH.
**Figure supplement 2—source data 1.** Quantification of FISH results (b).

Overexpression of RyhB or SgrS, in contrast, did not cause any significant changes in the Hfq-mMaple3 diffusivity or the corresponding mRNA-associated fraction (*Figure 3a and b*). We reasoned that there might be two possibilities. First, since class I sRNAs bind through the proximal face and the rim of Hfq, it can bind to the mRNA-free Hfq or co-occupy the mRNA-associated Hfq to generate sRNA-associated Hfq or sRNA–mRNA–Hfq ternary complex, respectively. Second, class I sRNAs cannot effectively compete against mRNAs for Hfq binding; therefore, most Hfq proteins remain associated with mRNAs. To distinguish these two possibilities, we examined the abundance of RyhB and SgrS compared to ChiX. Since the stabilities of RyhB and SgrS are highly dependent on Hfq (*Massé et al., 2003*), if the second hypothesis is correct, then we would expect a much lower cellular level of RyhB and SgrS compared to ChiX. We performed droplet digital PCR (ddPCR) in the same conditions as the tracking assays, and the result showed that RyhB or SgrS level was comparable to ChiX (*Figure 3d*). It should be noted that while ChiX level was almost fivefold of the RyhB level when normalized to the reads of 16S rRNA (*Figure 3d*, upper panel), ChiX level was about 50% of RyhB level when normalized to the reads from empty vector (representing the induction fold change) (*Figure 3d*, lower panel). We, therefore, reasoned that the difference between ChiX and RyhB when normalizing to the 16S rRNA was very likely due to the different efficiency during reverse transcription and PCR steps for these two targets. SgrS level was higher than ChiX with both normalizations. The observation supports that the stability of RyhB or SgrS is not compromised even though they do not displace mRNAs from Hfq, and therefore the first possibility that they either occupy the free Hfq or co-occupy on Hfq with the mRNA is more likely.

To further corroborate the observation that SgrS and RyhB can co-occupy Hfq with non-target mRNA, we performed the same EMSA competition assay using RyhB as an example (*Figure 3f*). When chasing with increasing concentration of RyhB, the band intensity of *ptsG*-Hfq complex decreased with an increased intensity of free *ptsG* and the appearance of an additional upper-shifted band that did not appear when chasing with ChiX (*Figure 3f*, red arrow). This result supports the possibility of the RyhB-*ptsG*-Hfq ternary complex formation. In the EMSA assay, we also observed direct displacement of *ptsG* fragment by RyhB, albeit less efficiently than by ChiX, which was not indicated by the in vivo imaging results. The exact cause of the discrepancy is unclear, but we speculate that it could result from the differences between the cellular conditions and in vitro setting. Nevertheless, the EMSA results still support that RyhB can have different mechanisms to gain access to mRNA-occupied Hfq, and that it is structurally possible to have RyhB co-occupy with a non-target mRNA on Hfq.

To summarize, our results collectively suggest that representative sRNAs for both class I and class II sRNAs can access mRNA-occupied Hfq in vivo. It is possible that the mechanisms can be generalized to other members of the two sRNA classes, that is, class I sRNAs can co-occupy the Hfq protein with mRNAs through different binding sites, whereas class II sRNAs can directly compete against the mRNAs at the distal face. Interestingly, fluorescence in situ hybridization (FISH) showed a much stronger signal for RyhB compared to ChiX (*Figure 3—figure supplement 2*), even though their levels were similar as revealed by ddPCR (*Figure 3d*). The weaker hybridization signal for ChiX is very likely a reflection of the larger protected region by Hfq on both distal and proximal faces, hindering FISH probe binding.

## Class II sRNAs require interaction with the proximal face and a strong AAN motif to compete for Hfq binding

We next sought to understand the molecular features that made ChiX a strong competitor for Hfq binding. When overexpressed in the *hfq* Q8A-*mMaple3* background (proximal face mutation), ChiX lost its capability to displace mRNAs from the mutant Hfq (*Figure 3a and b*), suggesting that additional binding affinity provided by the proximal face of Hfq is critical for displacing other RNAs from the distal face. *E. coli* Hfq prefers an (A-A-N)$_n$ sequence for distal face binding, where N can be any

nucleotide, and each monomer binds to one A–A–N repeat (*Robinson et al., 2014*). ChiX contains four AAN motifs (*Figure 3c*). We tested the effect of AAN motifs on conferring the competitive binding to Hfq over mRNAs. We generated and overexpressed ChiX mutants with one or two AAN motif(s) deleted (*Figure 3c*) and found that the fraction of remaining mRNA-associated Hfq increased, when the number of AAN motifs decreased (*Figure 3b and c*). Notably, the levels of WT ChiX in the *hfq* Q8A-*mMaple3* background, and the ChiX mutants in the WT *hfq-mMaple3* background remained similar as the WT ChiX in WT *hfq-mMaple3* background (*Figure 3d*), indicating that the observed difference was not due to a change in the cellular ChiX level.

## Hfq is deficient in releasing mRNAs without interactions with RNase E

The C-terminal region of RNase E serves as a scaffold for the degradosome protein components (RNA helicase RhlB, enolase, and polynucleotide phosphorylase [PNPase]). Hfq has been demonstrated to interact with the C-terminal scaffold region of RNase E, although it is still under debate whether such interaction is direct or mediated by RNA (*Bruce et al., 2018*; *Ikeda et al., 2011*; *Morita et al., 2005*; *Worrall et al., 2008*). To study whether the interaction with RNase E affects the diffusivity of Hfq, we imaged Hfq-mMaple3 in two RNase E mutant strains. The *rne131* mutant strain has RNase E truncated by the last 477 amino acid residues (*Lopez et al., 1999*); therefore, while it maintains its nuclease activity, this mutant cannot interact with Hfq. The *rneΔ14* mutant has a smaller fraction of the C-terminal scaffold (residues 636–845) deleted, encompassing the Hfq, RhlB, and enolase binding regions and two RNA-binding domains (*Leroy et al., 2002*).

In both RNase E mutant backgrounds, the diffusivity of WT Hfq-mMaple3 became less sensitive to transcription inhibition by rifampicin compared to the WT *rne* background (*Figure 4a*). For WT Hfq-mMaple3, ~75% of Hfq-mMaple3 became mRNA-free upon rifampicin treatment in the RNase E mutant backgrounds compared to ~89% of mRNA-free Hfq in the WT *rne* background (*Figure 4b*). Hfq Y25D-mMaple3 (the distal face mutant), which is deficient in mRNA binding, showed minimal sensitivity to rifampicin treatment in the *rne* mutant backgrounds, the same as in the WT *rne* background (*Figure 4a and b*). These observations suggest that without the Hfq-RNase E interaction, more mRNAs remained bound to Hfq, and hint that Hfq-RNase E interaction may help recycle Hfq from the mRNA-associated form through degradation of mRNAs. To investigate whether the increased mRNA-associated form in the absence of Hfq-RNase E interaction is primarily contributed

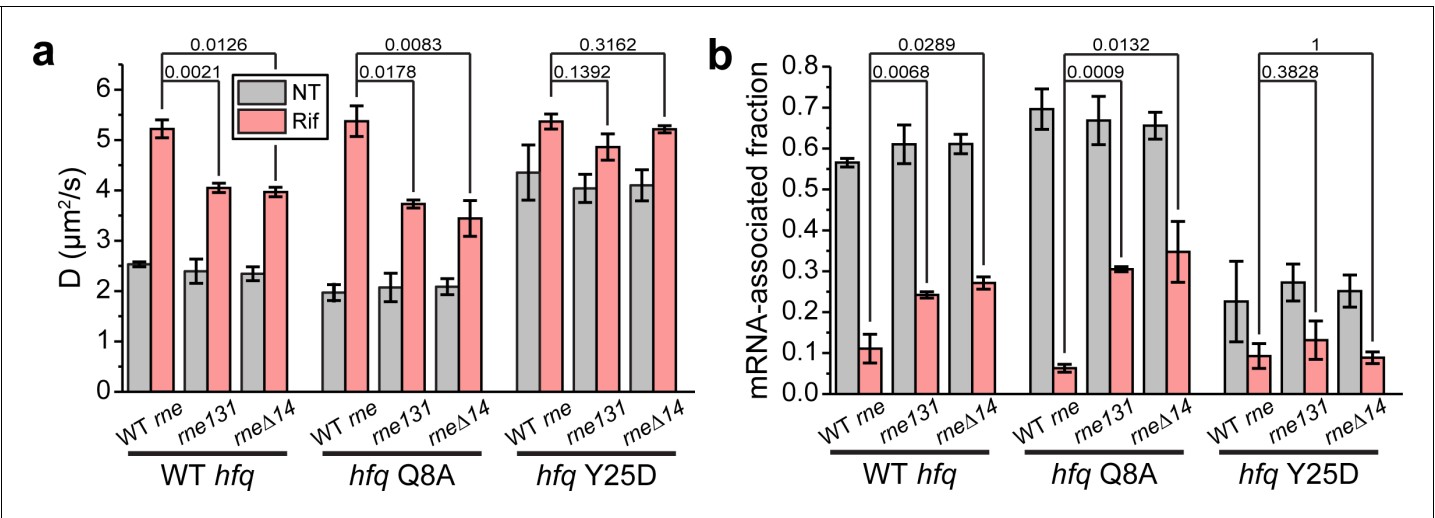

**Figure 4.** Hfq-RNase E interaction contributes to the recycling of Hfq from the mRNA-associated form to the mRNA-free form. (a) Diffusion coefficients are plotted for WT, Q8A, and Y25D Hfq-mMaple3 in the *rne131* and *rneΔ14* backgrounds under NT and Rif conditions. (b) mRNA-associated fraction of Hfq, for the same cases in (a). p-values are reported between WT *rne* and *rne* mutants under Rif condition. Error bars in all plots represent the s.d. from two experimental replicates, with each data set containing ~5,000 trajectories (for D value calculation) or ~20,000 trajectories (for mRNA-associated fraction calculation) from ~100 cells. All fitting results are reported in *Supplementary file 1*.

The online version of this article includes the following source data for figure 4:

**Source data 1.** Diffusion coefficients and mRNA-bound fractions of Hfq in the backgrounds of RNase E mutants.

by the regulation of sRNAs, we imaged Hfq Q8A-mMaple3 in the *rne* mutant backgrounds. Q8A mutation in the proximal face of Hfq broadly disrupts the binding and stabilization of sRNAs (*Schu et al., 2015*). If sRNA-dependent regulation is the sole pathway in changing Hfq from the mRNA-associated form to the mRNA-free form, we would expect a great increase in the mRNA-associated fraction of the Q8A mutant in the *rne* mutant backgrounds compared to the WT Hfq with rifampicin treatment. However, we observed a minor difference in the mRNA-associated fraction between Q8A mutant (~33% considering both *rne131* and *rneΔ14* backgrounds) and the WT Hfq (~25%) (*Figure 4b*), suggesting that the recycling of Hfq from the mRNA-associated form to the mRNA-free form can be affected by Hfq-RNase E interaction in addition to the sRNA-dependent regulatory pathway.

## Hfq-RNase E interaction contributes to the degradation of Hfq-associated mRNAs

As our results above indicate that Hfq-RNase E interaction contributes to the recycling Hfq from the mRNA-associated form, likely through degradation, we hypothesized that Hfq-RNase E interaction might play a role in the turnover of specific Hfq-bound mRNAs. To test this hypothesis, we used northern blots to measure the half-lives of selected mRNAs that are known to interact with Hfq in four backgrounds: (1) WT *hfq-mMaple3* + WT *rne*, (2) WT *hfq-mMaple3* + *rne131* mutant, (3) *hfq* Y25D-*mMaple3* (distal face mutant) + WT *rne*, and (4) *hfq* Y25D-*mMaple3* + *rne131* mutant (*Figure 5*). To test whether such regulation can occur in the absence of the corresponding sRNAs, in addition to the genetic background of *hfq-mMaple3* and *rne*, we also knocked out the corresponding sRNA regulators of the selected mRNAs (*ΔryhBΔfnrS* for *sodB*, *ΔcyaRΔmicA* for *ompX* and *ΔryhBΔspfΔrybB* for *sdhC*; herein simplified as 'ΔsRNA'). The choice of knocked-out sRNAs covers all sRNAs identified in a global mapping of sRNA-target interactions for the selected mRNAs in log phase (*Melamed et al., 2016*). In the WT *rne* background, these three mRNAs showed 47% to 2.2-fold increase in the half-lives in the *hfq* Y25D-*mMaple3* background compared to WT *hfq-mMaple3* (*Figure 5d*). In the *rne131* mutant background, while all mRNAs showed increased half-lives of 1.4 to 3.7-fold compared to the WT *rne* background, consistent with a compromised activity in the *rne131* mutant (*Lopez et al., 1999*), the differences in the mRNA half-lives between WT *hfq-mMaple3* and *hfq* Y25D-*mMaple3* backgrounds were largely diminished (*Figure 5e*). This result indicates that in the absence of Hfq-RNase E interaction, association with Hfq or not does not change the mRNA turnover.

To further exclude the contributions by potentially unknown sRNAs, we compared the lifetime of *sodB*, *ompX*, and *sdhC* in the *hfq* Q8A-*mMaple3* background in addition to knocking out corresponding sRNAs (*Figure 5*). The half-lives of these mRNAs increased by 16–45% in the *hfq* Q8A background compared to WT *hfq* background, smaller than the increase observed in the *hfq* Y25D background (*Figure 5d*). The increase of mRNA half-life in the *hfq* Q8A background can either be due to contributions by unknown sRNA regulators or due to other possible regulatory pathways by Hfq through binding at the proximal face. One of such regulatory pathways may be Hfq-mediated polyadenylation, which involves binding of Hfq at the Rho-independent termination site and promotes mRNA degradation (*Hajnsdorf and Régnier, 2000*; *Mohanty et al., 2004*). Despite these two possibilities, the increase in the mRNA half-life due to Y25D mutation cannot be fully explained by sRNA-mediated regulation. These results collectively support that besides the sRNA-mediated pathway, Hfq can facilitate the turnover of certain mRNAs by binding to the mRNAs through the distal face and bridging them to RNase E for degradation.

## RppH is not required in this Hfq-mediated mRNA turnover pathway

RNase E has two mechanisms for substrate recognition and subsequent endonuclease cleavage. RNase E can directly access and cleave RNA substrates with certain sequence preferences (*Chao et al., 2017*; *Clarke et al., 2014*). Alternatively, RNase E can recognize the 5′ monophosphate on RNA substrates for catalytic activation (*Mackie, 1998*; *Jiang and Belasco, 2004*; *Bandyra et al., 2012*). In this 5′-end dependent mechanism, RppH, a pyrophosphohydrolase, is needed to convert the 5′ triphosphate of the RNA substrate to 5′ monophosphate (*Celesnik et al., 2007*; *Deana et al., 2008*). To test whether this Hfq-mediated mRNA turnover is dependent on the 5′-end decapping,

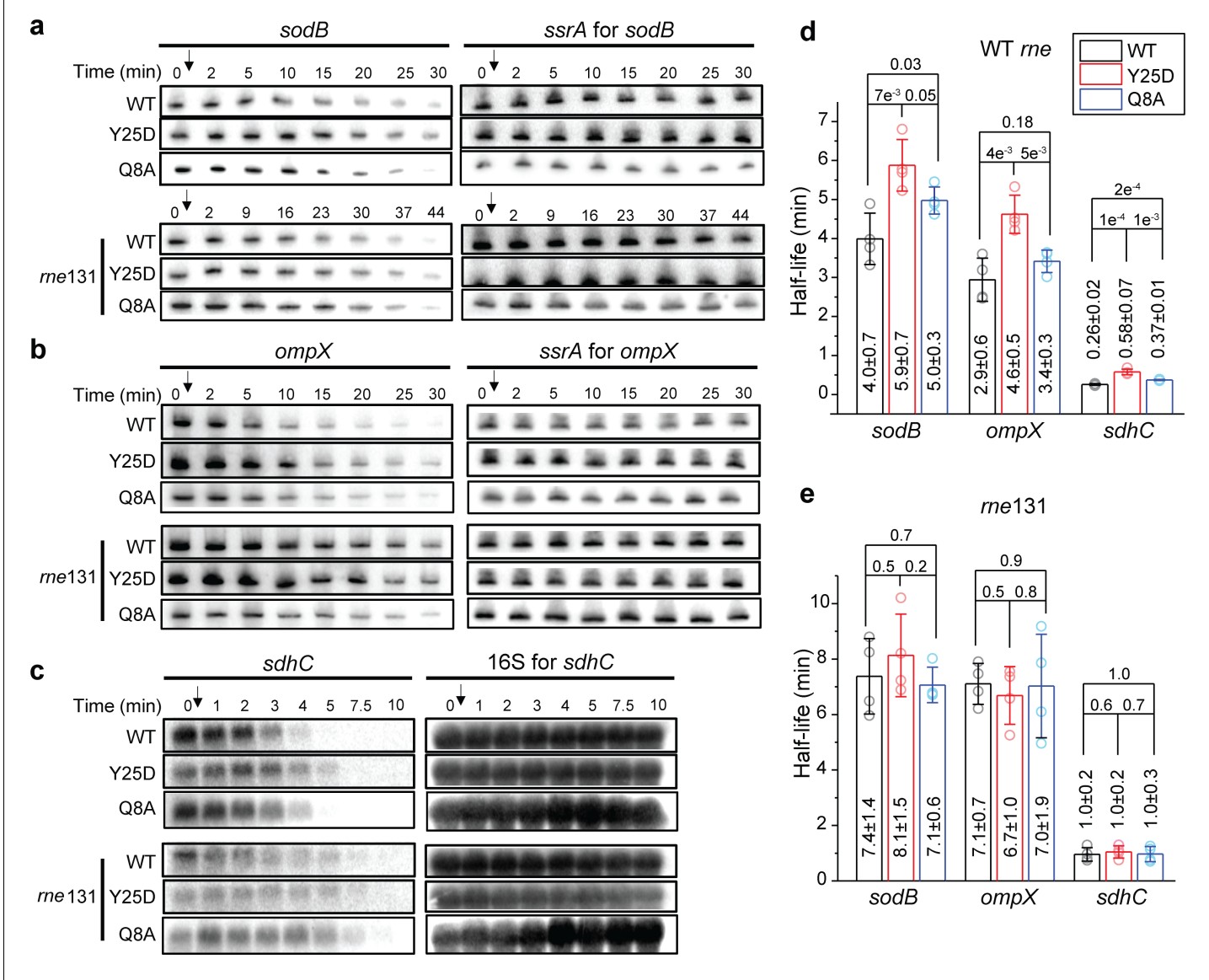

**Figure 5.** Hfq-RNase E interaction contributes to the regulation of mRNA degradation. The abundance of (**a**) *sodB*, (**b**) *ompX*, and (**c**) *sdhC* mRNA in the presence of WT Hfq, Hfq Y25D, or Hfq Q8A in the WT *rne* or rne*131* background. Corresponding sRNAs were knocked out for each of the mRNAs (Δ*ryhB*Δ*fnrS* for *sodB*, Δ*cyaR*Δ*micA* for *ompX*, and Δ*ryhB*Δ*spf*Δ*rybB* for *sdhC*). Strains were grown in MOPS EZ-rich medium containing 0.2% glucose until OD$_{600}$ = 0.5. Rifampicin was added as indicated by the arrow and total RNA was extracted at specific time points. *ssrA*, or 16S rRNA was used as loading controls. Relative abundance of mRNA quantified by densitometry as a function of time is presented in *Figure 5—figure supplement 1*. (**d** and **e**) Half-lives of the mRNAs determined from (**a**) to (**c**). Scatter plot represents the data points from individual replicates, and bar graph with error bars represents the mean and s.d. of four biological replicates. p-values from t-test are reported for each pairwise comparison.

The online version of this article includes the following source data and figure supplement(s) for figure 5:

**Source data 1.** Decay rates and half-lives of mRNAs.

**Figure supplement 1.** Quantification of northern blot.

we used *sdhC* as an example, and compared its half-life in the backgrounds of ΔsRNAΔ*rppH* and ΔsRNAΔ*rppH hfq* Y25D.

In the ΔsRNAΔ*rppH* background, *sdhC*'s half-life was ~2.6-fold of the half-life in the ΔsRNA background, suggesting that the action of RppH contributes to the endogenous turnover rate of *sdhC* in general (*Figures 5d* and *6a, b*; *Deana et al., 2008*). However, in the ΔsRNAΔ*rppH* background, additional Y25D mutation on Hfq caused ~90% increase in the half-life compared to that in the

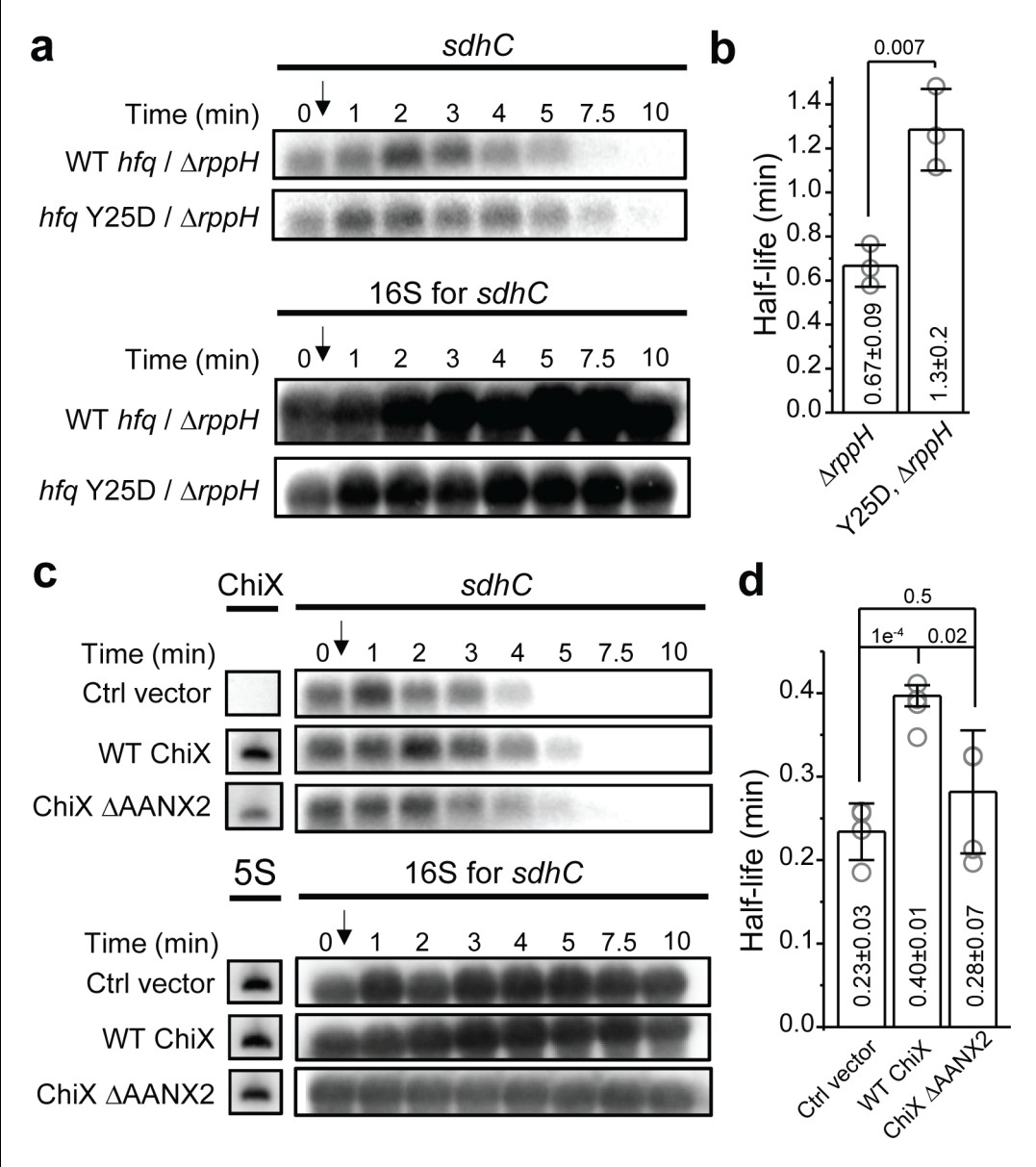

**Figure 6.** Effect of RppH and ChiX on Hfq-mediated regulation on mRNA degradation. (**a**) The abundance of *sdhC* mRNA in the presence of WT Hfq or Hfq Y25D in the Δ*ryhB*Δ*spf*Δ*rybB*Δ*rppH* background. Strains were grown in MOPS EZ-rich medium containing 0.2% glucose and rifampicin was added at $OD_{600}$ = 0.5. (**b**) Half-life of *sdhC* mRNA determined from (**a**). (**c**) The abundance of *sdhC* mRNA in the presence of control vector, WT ChiX, and mutant ChiX with two AAN motif deleted. Strains were grown in MOPS EZ-rich medium containing 0.2% fructose and 1 mM IPTG was added at $OD_{600}$ = 0.1 to induce ChiX for 1 hr before addition of rifampicin. (**d**) Half-life of *sdhC* mRNA determined from (**c**). Scatter plot represents the data points from individual replicates, and bar graph with error bars represents the mean and s.d. of three to four biological replicates. p-values from t-test are reported for each pairwise comparison.

The online version of this article includes the following source data and figure supplement(s) for figure 6:

**Source data 1.** Decay rates and half-lives of mRNAs.

**Figure supplement 1.** Quantification of northern blot.

presence of WT Hfq (*Figure 6a and b*). The half-life increase caused by Hfq Y25D mutation in the Δ*rppH* background was comparable with the half-life increase in the WT *rppH* background (120% increase in half-life in the background of ΔsRNA *hfq* Y25D compared to ΔsRNA [*Figure 5d*] for *sdhC*). These results suggest that the decapping by RppH is not required for the Hfq-mediated regulation of mRNA turnover, at least for the case of Hfq regulation on *sdhC* mRNA.

### sRNA that competes for Hfq binding can modulate Hfq's ability to regulate mRNA turnover

As our model suggests that binding of Hfq to the mRNA through the distal face can regulate the mRNA turnover, we reasoned that sRNAs that can effectively compete for Hfq binding against mRNAs may decoy Hfq from this regulatory function. To test this, we again used *sdhC* as an example and measured its half-life in the presence of ChiX, which is a strong competitor for Hfq binding (*Figure 3*). In the presence of vector control, *sdhC* exhibited comparable half-life compared to the case without any plasmid (*Figures 5d* and *6c, d*). The presence of WT ChiX increased the half-life by ~70%, whereas the mutant ChiX without two AAN motif did not cause a significant increase in the half-life of *sdhC*, consistent with its reduced binding ability to Hfq (*Figures 3a, b*, *6c and d*). These results further support our model of Hfq-mediated regulation of mRNA turnover and demonstrate that the presence of strong Hfq binding sRNAs can modulate the strength of Hfq's regulation.

## Discussion

Using single-particle tracking, we resolved different diffusivity states of Hfq proteins in live cells, reporting on the interactions with different cellular RNAs. Specifically, free Hfq and sRNA-associated Hfq proteins (collectively termed as 'mRNA-free Hfq') have a high diffusivity, and association of mRNAs to form mRNA-associated Hfq or sRNA-mRNA-Hfq ternary complex (collectively termed as 'mRNA-associated Hfq') reduces the diffusivity of Hfq (*Figure 7a*). Our results are reminiscent of a previously proposed model of Hfq interacting with sRNAs and mRNAs in a face-dependent manner (*Schu et al., 2015*). During exponential growth, Hfq proteins are largely occupied by mRNAs. The distal face of Hfq is the primary binding site for cellular mRNAs, while the rim has a minor binding role. These observations suggest that the majority of the Hfq-bound mRNAs are class I mRNAs, and a minority are class II mRNAs, consistent with the previous findings that most of the sRNAs are class I sRNAs (*Schu et al., 2015*). Under the conditions when specific sRNAs are highly induced, both classes of sRNAs can easily access Hfq upon induction, albeit with different mechanisms (*Figure 7b*). Our data demonstrate that class II sRNAs, such as ChiX, can effectively displace class I mRNAs from

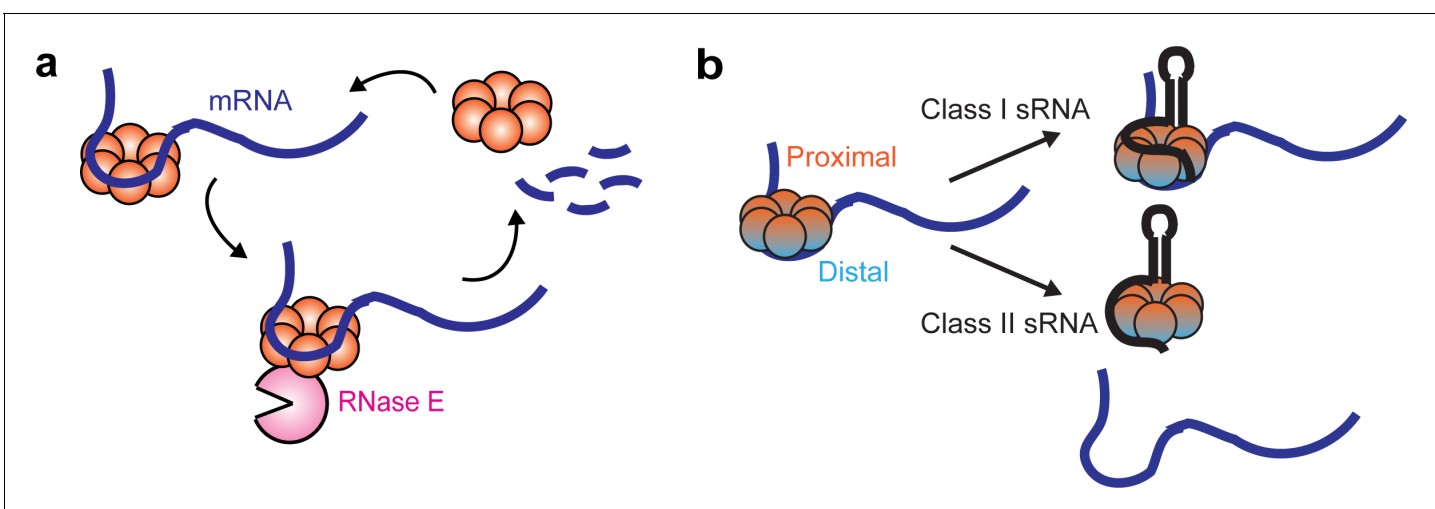

**Figure 7.** Dynamic interactions between Hfq and cellular RNAs. (**a**) Hfq facilitates the degradation of certain Hfq-bound mRNAs through the recruitment of RNase E. (**b**) Class I or Class II sRNAs can get access of mRNA-associated Hfq through co-occupying different binding sites of Hfq simultaneously, or displacing mRNA from the distal face of Hfq respectively.

the distal face, consistent with the proposed RNA exchange model. Interestingly, our data indicate that class I sRNAs do not necessarily need to displace mRNA from Hfq. Instead, they can directly co-occupy Hfq through the binding faces that are non-overlapping with the class I mRNA binding face. In both cases, the mRNA-associated Hfq proteins are in standby mode for sRNA binding if needed. The displacement of mRNA by the class II sRNA requires both the interactions at the proximal face of Hfq and higher AAN motif number to outcompete mRNAs for the binding at the distal face. In addition, we propose that the competitive binding by the class II sRNA is likely to occur stepwise, with binding at the proximal face happening first, followed by the displacement of mRNA from the distal face, which is supported by the observation that with the Hfq proximal face mutation, ChiX cannot displace mRNAs, even with a strong AAN motif.

We observed from live-cell tracking experiments that recycling of Hfq from the mRNA-associated form to the mRNA-free form upon rifampicin treatment is compromised in the RNase E mutant backgrounds, where regions including the Hfq binding site are deleted. This observation suggests that RNase E can facilitate the recycling of Hfq from the mRNA-associated form through mRNA degradation. Additional half-life measurements on a few selected mRNAs under various genetic backgrounds further demonstrate that Hfq can facilitate the turnover of certain mRNAs through binding with its distal face and recruiting RNase E, and this regulation is most likely to be independent of their corresponding sRNA regulators. Interestingly, we observed that sRNA competitors, such as ChiX, which can outcompete mRNAs for binding at the distal face, can decoy Hfq from regulating mRNA turnover, the same effect as the distal face Y25D mutation. Similar observation was reported previously that ChiX can titrate Hfq from translationally repressing transposase mRNA (*Ellis et al., 2015*). Considering our mutational work on ChiX-Hfq interaction, it should be possible to engineer synthetic sRNAs to tune Hfq-RNA interactions and Hfq regulatory functions in vivo.

Mechanisms of sRNA-independent Hfq-mediated regulation on mRNA turnover have been reported. First, binding Hfq, or Hfq in complex with other proteins such as Crc, at the ribosome binding site of the mRNA can repress translation (*Urban and Vogel, 2008*; *Vytvytska et al., 2000*; *Pei et al., 2019*), therefore indirectly increasing the mRNA degradation due to de-protection of the mRNA by translating ribosomes against RNase E. Since this translation-dependent regulation of Hfq does not require Hfq-RNase E interaction, if this mechanism applied to the mRNAs we tested, we would expect the difference in half-life between the cases of WT *hfq* and *hfq* Y25D to be similar in the WT *rne* and *rne131* backgrounds. The observation that the difference in half-life due to Hfq binding is eliminated in the *rne131* background suggests that the turnover of these mRNAs is not primarily through regulation at the translation level. Second, binding of Hfq may recruit polyA polymerase (PAP) and PNPase to stimulate polyadenylation at the 3′ end, and therefore promote degradation (*Hajnsdorf and Régnier, 2000*; *Mohanty et al., 2004*), an action that may also involve interactions with the C-terminal scaffold region of RNase E. However, this mechanism also cannot fully explain our results, as Hfq-stimulated polyadenylation prefers Hfq binding at the 3′ termini of mRNAs containing Rho-independent transcription terminator (*Mohanty et al., 2004*), whereas in our selected mRNAs, they do not all utilize Rho-independent termination, and the binding sites of Hfq on the tested mRNAs are within the 5′ UTR and CDS region containing A-rich motif, based on the CLIP-seq analysis of Hfq (*Tree et al., 2014*). Therefore, it is more likely that the regulation for the selected mRNAs is through recruitment of RNase E rather than through Hfq-stimulated polyadenylation mechanism. Nevertheless, we expect Hfq can potentially regulate mRNA turnover through a combination of these mechanisms in a gene-specific manner, which explains our observation that depending on the specific mRNA, Hfq can facilitate the turnover rate through distal face binding to different extent.

The endonuclease activity of RNase E has been shown to either be dependent on 5′-monophosphate of the RNA substrate, or independent, with the former mechanism requiring RppH to convert 5′-triphosphate cap to 5′-monophosphate. Using *sdhC* mRNA as an example, our results demonstrate that the 5′-monophosphate dependent pathway contributes insignificantly to this specific Hfq-mediated regulation of mRNA turnover. While our results reveal that Hfq binding contributes to mRNA turnover through recruitment of RNase E, more mechanistic details remain to be further elucidated. First, it is under debate whether Hfq-RNase E interaction is direct or mediated by RNAs (*Bruce et al., 2018*; *Ikeda et al., 2011*; *Morita et al., 2005*; *Worrall et al., 2008*). While our data show that Hfq can promote mRNA degradation without their matching regulatory sRNAs, it remains to be investigated whether other cellular RNAs may participate in bridging the interaction. Second,

while our data demonstrate that deletion of the scaffold region of RNase E abolishes the Hfq-mediated regulation on mRNA turnover, the same scaffold region also includes the binding sites of other degradomes components; thus we cannot exclude the possibility that these protein components also participate in the regulation. Further experiments are needed to answer these mechanistic questions.

# Materials and methods

## Key resources table

| Reagent type (species) or resource | Designation | Source or reference | Identifiers | Additional information |
|---|---|---|---|---|
| Chemical compound, drug | a-a'-dipyridyl crystalline | Fisher Scientific | | Catalog #: D95-5 |
| Chemical compound, drug | D-fructose | Bioshop | | Catalog #: FRC180 |
| Chemical compound, drug | IPTG (for in vitro study) | Bioshop | | Catalog #: IPT001 |
| Chemical compound, drug | IPTG (for microscopy imaging) | Goldbio | | 367-93-1 |
| Chemical compound, drug | L(+)-arabinose | Bioshop | | Catalog #: ARB222 |
| Chemical compound, drug | Rifampicin (for in vitro study) | Bioshop | | Catalog #: RIF222 |
| Chemical compound, drug | Rifampicin (for microscopy imaging) | Sigma-Aldrich | | R3501-1G |
| Commercial assay, kit | 2× QX200 ddPCR EvaGreen Supermix | Bio-Rad | RRID:SCR_019707 | Catalog #: 1864034 |
| Other | EZ-rich define media | Teknova | | Catalog #: M2105 Bacterial growth media |
| Other | HiTRAP Heparin column HP (resin) | GE Healthcare Life Sciences | | Catalog #: 17-0406-01 Protein purification column |
| Peptide, recombinant protein | Transcriptor reverse transcriptase | Roche | | Catalog #: 3531317001 |
| Software, algorithm | GraphPad Prism | GraphPad Prism | RRID:SCR_002798 | Version 8.2.1 |
| Software, algorithm | Image studio lite | Li-COR | RRID:SCR_013715 | Version 5.2 |
| Software, algorithm | MATLAB | MathWorks | | Version R2019b |
| Software, algorithm | NIS-Element | Nikon | | Version 4.51 |
| Software, algorithm | QuantaSoft | Bio-Rad | | Catalog #: 1864011 |

## Bacterial strains

Transfer of the Linker-mMaple3-Kan sequence at the 3' end of chromosomal *hfq* gene was achieved by following the PCR-based method (*Datsenko and Wanner, 2000*) with a few modifications. First, a PCR (PCR1) was performed using plasmid pZEA93M as template to amplify the mMaple3 sequence (oligos EM4314-4293). Then, to add sequence homology of *hfq* gene, a second PCR (PCR2) was performed using PCR1 product as template (oligos EM4313-4293). The final PCR product (PCR3), containing a flippase recognition target (FRT)-flanked kanamycin resistance cassette, was generated from pKD4 plasmid as template with PCR2 product and oligo EM1690 as primers carrying extensions homologous to the *hfq* gene. PCR3 was then purified and transformed to WT (EM1055), *hfq* Y25D (KK2562), *hfq* Q8A (KK2560), *hfq* K31A (AZZ41), or *hfq* R16A (KK2561) strains containing the pKD46 plasmid using electroporation, to obtain strains with *hfq*-Linker-mMaple3-Kan, *hfq* Y25D-Linker-mMaple3-Kan, *hfq* Q8A-Linker-mMaple3-Kan, *hfq* K31A-Linker-mMaple3-Kan, and *hfq* R16A-Linker-mMaple3-Kan, respectively. Hfq mutations F42A and R19D were obtained by performing PCRs on fusion strains KP1867 (Hfq-linker-mMaple3) with oligos EM4704-1690 (Hfq F42A) or EM4705-1690 (Hfq R19D). Fragments were then transformed to WT (EM1055) containing the pKD46 plasmid, following induction of the λ Red. P1 transduction was used to transfer the linked FP and the antibiotic resistance gene into a WT (EM1055), *rne131* (EM1377), or *rne*Δ14 (EM1376) strains. *fnrS*, *micA*, *spf*, *chiX*, and *rppH* knockouts were obtained through transformation of PCR products into EM1237 after induction of λ *red* and selecting for kanamycin or chloramphenicol resistance. P1 transduction was used to transfer the knockout mutations and the antibiotic resistance gene into appropriate strains. Selection was achieved with kanamycin, chloramphenicol, or tetracycline. When necessary, FRT-flanked antibiotic resistance cassettes were eliminated after transformation with pCP20, as described (*Datsenko and Wanner, 2000*). P1 transduction was also used to transfer *hfq*-

Linker-mMaple3-Kan, *hfq* Y25D-Linker-mMaple3-Kan and *hfq* Q8A-Linker-mMaple3-Kan in the ΔsRNA or *rne131*-ΔsRNA strains. All constructs were verified by sequencing and are listed in *Supplementary file 2*. Oligonucleotides used for generating constructs are listed in *Supplementary file 3*.

## Plasmids

*E. coli* MG1655 sRNA genes *sgrS*, *ryhB*, and *chiX* were inserted into the pET15b vector (kind gift from Perozo lab) to create plasmids pET15b-RyhB, pET15b-SgrS, and pET15b-ChiX using Gibson Assembly (in house) using oligos listed in *Supplementary file 2*. The *chiX ΔAAN* mutants were made using site directed mutagenesis. Primers EH159, EH160, and EH161 homologous to *chiX* while excluding the AAN domain were used to amplify the plasmid. The products were phosphorylated (NEB M0201S) and ligated (NEB M0202S) before transformation.

Cloning of pBAD-*micA* was performed by PCR amplification of *micA* (oligos EM2651-2652) on WT strain (EM1055). The PCR product was digested with SphI and cloned into a pNM12 vector digested with MscI and SphI. Two DNA fragments encoding the Fab fragment of the sAB-70 synthetic antibody were PCR amplified from pRH2.2-70-4D5-EA plasmid with OSA996/OSA997 and OSA998/OSA999 primer pairs. The third fragment (plasmid backbone) was amplified with OSA992/OSA993 primer pair from pZESA93M plasmid, and these three DNA fragments were joined to construct pZEFabM with RAIR assembly (*Watson and García-Nafría, 2019*). Construction of pZEGCN4MSD plasmid encoding the scFv-GCN4 binding protein was performed by amplifying the scFv-GCN4 reading frame (with oligos OSA1010/OSA1011) from pHR-scFv-GCN4-sfGFP-GB1-NLS-dWPRE plasmid. The plasmid backbone was PCR amplified with OSA1008/OSA1009 DNA oligos using pZESA93M as a template. The fragments were joined into pZEGCN4MSD with RAIR assembly. All constructs were verified by sequencing and are listed in *Supplementary file 2*. Oligos used for generating the constructs are listed in *Supplementary file 3*.

## Growth conditions for imaging experiments

Overnight cultures of *E. coli* strains were diluted by 1:100 in MOPS EZ-rich defined medium (Teknova). 0.2% glucose was used as the carbon source for imaging Hfq-mMaple3 WT and mutants under NT and rifampicin treated conditions. 0.2% fructose was used as the carbon source with 100 µg/mL ampicillin for cases with sRNA overexpression, mMaple3 control, mMaple3 fused sAB-70, and mMaple3 fused scFv-GCN4. Cultures were grown at 37°C aerobically. Plasmid-encoded sRNAs were induced by 1 mM IPTG when the $OD_{600}$ of the cell culture was ~0.1. Induced cells were grown for ~45 min before imaging. Plasmid-encoded mMaple3 protein (with 100–400 µM IPTG), mMaple3 fused sAB-70 (with 1 mM IPTG), and mMaple3 fused scFv-GCN4 (with 1 mM IPTG) were expressed and imaged in the same way. For the rifampicin treatment, rifampicin was added to a final concentration of 200 µg/mL when the $OD_{600}$ of the cell culture was ~0.2, and the cells were incubated for 15 min before imaging.

## Growth curve measurement

The bacterial strains were grown overnight in LB or MOPS EZ-rich medium containing 0.2% glucose. Cultures were diluted to $6 \times 10^6$ cells/mL in their respective medium and samples were prepared in triplicate by mixing 50 µL of cells and 50 µL of fresh medium to obtain $3 \times 10^6$ cells/mL. Assay was performed in Microtest plate, 96-well, flat base, polystyrene, sterile (Sarstedt) and growth was monitored using Epoch 2 Microplate Spectrophotometer reader (BioTek) with the following settings: OD = 600 nm, Temperature = 37°C, Reading = every 10 min for 22 hr, Continuous shaking.

## RNA extraction and northern blot analysis

Total RNA was extracted following the hot-phenol protocol as described (*Aiba et al., 1981*). To test the function of the mMaple3-tagged Hfq and compare that with the WT Hfq, cells were grown in LB to the $OD_{600}$ of 0.5 and either RyhB was induced by adding of 2.2′-dipyridyl in a WT *hfq* or in an *hfq-mMaple3* background, or MicA was induced by addition of 0.1% arabinose (ara) in a Δ*micA* WT *hfq* or in a Δ*micA hfq-mMaple3* background (pBAD-*micA*).

Determination of RNA half-life was performed in MOPS EZ-rich defined medium (Teknova) with 0.2% glucose by addition of 500 µg/mL rifampicin to the culture at the $OD_{600}$ of 0.5 before total

RNA extraction. Northern blots were performed as described previously (*Desnoyers and Massé, 2012*) with some modifications. Following total RNA extraction, 5–10 µg of total RNA was loaded on polyacrylamide gel (5% acrylamide 29:1, 8 M urea) and 20 µg was loaded on agarose gel (1%, 1× MOPS). Radiolabeled DNA and RNA probes used in this study are described in *Supplementary file 3*. The radiolabeled RNA probes used for northern blot analysis were transcribed with T7 RNA polymerase from a PCR product to generate the antisense transcript of the gene of interest (*Desnoyers et al., 2009*). Membranes were then exposed to phosphor storage screens and analyzed using a Typhoon Trio (GE Healthcare) instrument. Quantification was performed using the Image studio lite software (LI-COR).

The decay rate of mRNA degradation was calculated as previously described (*Moffitt et al., 2016*). Briefly, the intensity of northern blot at each time point upon adding rifampicin was normalized to the intensity at time zero and was fit by a piecewise function in the log space:

$$\ln I(t) = \begin{cases} \ln I(0), & t \leq \alpha \\ \ln I(0) - k(t-\alpha), & t > \alpha \end{cases}$$

where *I(t)* is the normalized intensity at time t, *I(0)* is the normalized intensity at time zero, *k* is the rate of exponential decay, and $\alpha$ is the duration of the initial delay before the exponential decay begins. The reported half-lives ($\tau$) are calculated by $\tau = \log(2)/k$ .

## Droplet digital PCR

Droplet digital PCR (ddPCR) was performed on total RNA extracted following the hot-phenol protocol (*Aiba et al., 1981*) from cells grown in MOPS EZ-rich defined medium containing 0.2% fructose (Teknova) with 50 µg/mL ampicillin. 1 mM IPTG was added at $OD_{600}$ = 0.1 for 1 hr before total RNA extraction. Samples were treated with 8 U Turbo DNase (Ambion) for 1 hr. RNA integrity was assessed with an Agilent 2100 Bioanalyzer (Agilent Technologies). Reverse transcription was performed on 1.5 µg total RNA with Transcriptor reverse transcriptase, random hexamers, dNTPs (Roche Diagnostics), and 10 U of RNase OUT (Invitrogen) following the manufacturer's protocol in a total volume of 10 µL.

ddPCR reactions were composed of 10 µL of 2× QX200 ddPCR EvaGreen Supermix (Bio-Rad), 10 ng (3 µL) cDNA, 100 nM final (2 µL) primer pair solutions, and 5 µL molecular grade sterile water (Wisent) for a 20 µL total reaction. Primers are listed in *Supplementary file 3*. Each reaction mix (20 µL) was converted to droplets with the QX200 droplet generator (Bio-Rad). Droplet-partitioned samples were then transferred to a 96-well plate, sealed, and cycled in a C1000 deep well Thermocycler (Bio-Rad) under the following cycling protocol: 95°C for 5 min (DNA polymerase activation), followed by 50 cycles of 95°C for 30 s (denaturation), 59°C for 1 min (annealing), and 72°C for 30 s (extension) followed by post-cycling steps of 4°C for 5 min and 90°C for 5 min (Signal stabilization) and an infinite 12°C hold. The cycled plate was then transferred and read using the QX200 reader (Bio-Rad) either the same or the following day post-cycling. The concentration reported is copies/µL of the final 1× ddPCR reaction (using QuantaSoft software from Bio-Rad) (*Taylor et al., 2015*).

## Hfq purification

Hfq was purified following the previously described procedure (*Prévost et al., 2007*) with modifications. Briefly, strain EM1392 containing pET21b-*hfq* was grown at 37°C in LB medium supplemented with 50 µg/mL ampicillin and 30 µg/mL chloramphenicol until it reached an $OD_{600}$ = 0.6. Hfq expression was induced by addition of 5 mM IPTG (Bioshop) for 3 hr. Cells were pelleted by centrifugation (15 min, 3825 g) and resuspended in 4 mL Buffer C (50 mM Tris-HCl pH 7.5, 1 mM EDTA, 50 mM $NH_4Cl$, 5% glycerol) (*Zhang et al., 2002*) supplemented with 30 U Turbo DNase (Ambion). Cells were lysed by sonication for 4 min (amplitude 25%, cycles of 5 s sonication, 5 s on ice) and samples were cleared by centrifugation (45 min, 12,000 g). The supernatant was incubated at 80°C for 10 min, centrifuged again (20 min, 12,000 g), and cleared by filtration.

The protein extract was loaded onto a 1 mL HiTRAP Heparin column HP (GE Healthcare Life Sciences, 17-0406-01) equilibrated with Buffer A (50 mM Tris-HCl pH 8.0, 50 mM NaCl, 50 mM KCl, 1 mM EDTA, 5% glycerol). After washes, the protein was eluted with a linear NaCl gradient (0.05–1 M NaCl) in Buffer A. Fraction samples were loaded on SDS-PAGE and stained with Coomassie-Blue. Hfq-containing fractions were dialyzed against a dialysis buffer (50 mM Tris-Cl pH 7.5, 1 mM EDTA

pH 8.0, 5% Glycerol, 0.25 M NH$_4$Cl). Glycerol concentration was brought up to 10% and protein content was quantified by BCA assay (Thermo Scientific).

## EMSA

DNA templates containing a T7 promoter were synthesized by PCR amplification on genomic DNA using oligonucleotides EM88-EM1978 (T7-*ryhB*), T7-ChiX(F)-T7-ChiX(R) (T7-*chiX*), or T7-ptsG(F)-T7-ptsG(R) (T7-*ptsG*). Briefly, templates were incubated for 4 hr at 37°C in RNA Transcription Buffer (80 mM HEPES-KOH pH 7.5, 24 mM MgCl$_2$, 40 mM DTT, 2 mM spermidine) in the presence of 5 mM NTP, 40 U porcine RNase Inhibitor (in house), 1 μg pyrophosphatase (Roche), and 10 μg purified T7 RNA polymerase (in house). Samples were treated with 2 U Turbo DNase (Ambion) and purified on polyacrylamide gel (6% acrylamide:bisacrylamide 19:1, 8 M urea). When necessary, transcripts were dephosphorylated using 10 U Calf Intestinal Phosphatase (NEB) and were 5' end-radiolabeled with [γ-$^{32}$P]-ATP using 10 U T4 polynucleotide kinase (NEB). Radiolabeled transcripts were purified on polyacrylamide gel (6% acrylamide:bisacrylamide 19:1, 8 M urea).

EMSA were performed as previously described (*Morita et al., 2012*). To determine binding affinity of Hfq to RyhB, ChiX, and *ptsG*, radiolabeled RNA was heated for 1 min at 90°C and put on ice for 1 min. RNA was diluted to 20 nM in modified Binding Buffer 2 (10 nM Tris-HCl pH 8.0, 1 mM DTT, 1 mM MgCl$_2$, 20 mM KCl, 10 mM Na$_2$HPO$_4$-NaH$_2$PO$_4$ pH 8.0, 12.5 μg/mL yeast tRNA) and mixed with specific concentrations of Hfq (0–200 nM). Samples were incubated for 15 min at 37°C and reactions were stopped by addition of 1 μL of non-denaturing loading buffer (1× TBE, 50% glycerol, 0.1% bromophenol blue, 0.1% xylene cyanol). For competition assays, 20 nM of radiolabeled *ptsG* was first incubated for 15 min at 37°C with 100 nM Hfq (as described above). Specific concentrations of RyhB or ChiX (0–100 nM) were added to the samples and incubation was carried out for 15 min at 37°C. Reactions were stopped by addition of 1 μL non-denaturing loading buffer. Samples were loaded on native polyacrylamide gels (5% acrylamide:bisacrylamide 29:1) in cold TBE 1X and migrated at 50 V, at 4°C. Gels were dried and exposed to phosphor storage screens and analyzed using a Typhoon Trio (GE Healthcare) instrument. When applicable, quantification was performed using the Image studio lite software (LI-COR) and data was fitted using nonlinear regression (GraphPad Prism).

## Fluorescence in situ hybridization

Sample preparation for fixed cells was performed mostly according to the protocol previously reported (*Park et al., 2018a*; *Fei et al., 2015*). Briefly, ~10 mL of cell culture was collected and fixed with 4% formaldehyde in 1× PBS for 30 min at room temperature (RT). The fixed cells were then permeabilized with 70% ethanol for 1 hr at RT. Permeabilized cells can be stored in 70% ethanol at 4°C until the sample preparation. FISH probes were designed and dye-labeled as in the previous reports (*Park et al., 2018a*; *Fei et al., 2015*). Hybridization was performed in 20 μL of hybridization buffer (10% dextran sulfate [Sigma D8906] and 10% formamide in 2× SSC) containing specific sets of FISH probes at 30°C in the dark overnight. The concentration of FISH probes was 50 nM. After hybridization, cells were washed three times with 10% FISH wash buffer (10% formamide in 2× SSC) at 30°C.

## Live-cell single-particle tracking and fixed cell SMLM imaging

Imaging was performed on a custom built microscopy setup as previously described (*Park et al., 2018b*). Briefly, an inverted optical microscope (Nikon Ti-E with 100× NA 1.49 CFI HP TIRF oil immersion objective) was fiber-coupled with a 647 nm laser (Cobolt 06–01), a 561 nm laser (Coherent Obis LS) and a 405 nm laser (Crystalaser). A common dichroic mirror (Chroma zt405/488/561/647/752r-UF3) was used for all lasers, but different emission filters were used for different fluorophores (Chroma ET700/75M for Alexa Fluor 647 and Chroma ET595/50M for mMaple3). For imaging Hoechst dye, a LED lamp (X-Cite 120LED) was coupled with a filter cube (Chroma 49000). The emission signal was captured by an EMCCD camera (Andor iXon Ultra 888) with slits (Cairn OptoSplit III), enabling fast frame rates by cropping the imaging region. During imaging acquisition, the Z-drift was prevented in real time by a built-in focus lock system (Nikon Perfect Focus).

For live-cell single particle tracking, 1 mL of cell culture was centrifuged at 1500 g for 5 min and 970 μL of the supernatant was removed. The remaining volume was mixed well and ~1.5 μL was covered by a thin piece of 1% agarose gel on an ethanol-cleaned-and-flamed coverslip sealed to a

custom 3D printed chamber. The agarose gel contained the same concentration of any drug or inducer used in each condition. Exceptions include rifampicin, which was at 100 µg/mL in the gel, due to high imaging background caused by high concentration of rifampicin, and IPTG for mMaple3 alone control culture, which was eliminated in the gel, due to the high abundance mMaple3 already induced by IPTG in the culture. The power density of the 561 nm laser for single-particle tracking was ~2750 W/cm$^2$, and the power density of the 405 nm laser was ~7 W/cm$^2$ (except for mMaple3 alone control where ~4.5 W/cm$^2$ was used due to high abundance of mMaple3). 1.5× Tube lens was used for the microscope body, and 2 × 2 binning mode was used for the camera. In this way, the effective pixel size became larger (173 nm instead of original 130 nm), receiving 77% more photons per pixel. Ten frames with 561 nm excitation were taken after each frame of 405 nm photo-conversion. About 13,000 frames were collected per movie at a rate of 174 frames per second. For fixed-cell control experiment for tracking parameter optimization, imaging was performed using the exact same imaging parameters as in live-cell measurements for a fair comparison. In cases imaging DNA together, Hoechst dye (Thermo 62249) was added to the ~30 µL of cell culture before imaging at ~20 µM final concentration and imaged by the LED lamp (12%) with 500 ms exposure time. Imaging acquisition was conducted by NIS-Element (Nikon) software, at RT.

## Image reconstruction

The SMLM images are reconstructed as previously descried (*Fei et al., 2015*), by a custom code written in IDL (Interactive Data Language). Briefly, all the pixels with an intensity value above the threshold were identified in each frame. The threshold was set at three times of the standard deviation of the individual frame pixel intensity. Among those pixels, the ones having larger values than surrounding pixels in each 5 × 5 pixel region were identified as possible peak candidates, and 2D Gaussian function was fit to a 7 × 7 pixel region surrounding these candidates. Candidates with failed fitting were discarded, and precise peak positions were defined for the remaining ones. The horizontal drift, which often occurred during the imaging acquisition, was corrected by fast Fourier transformation analysis.

## Tracking analysis

We used a MATLAB coded tracking algorithm to generate diffusion trajectories, which was modified by Sadoon and Yong (*Sadoon and Wang, 2018*) based on the previously developed code (*Crocker and Grier, 1996*). Per each time step of ~5.76 ms, 400 nm was empirically chosen to be the maximum one-step displacement to reduce artificial diffusion trajectories connected between different molecules, using a fixed cell sample as a control (*Figure 1—figure supplement 3B*). Trajectories longer than five time steps were used to calculate effective diffusion coefficient (*D*). MSD as a function of time lag (Δ*t*) was fit with a linear function (MSD = *D* × Δ*t*). *D* values are reported in related figures. For analysis using one-step displacement (*osd*), trajectories longer than three time steps were used.

## Enrichment calculation

Enrichment at a certain region (nucleoid, membrane, or cytoplasm) of a cell is defined as follows:

$$\frac{(\# \, of \, localizations \, in \, the \, region)/(total \, \# \, of \, localizations \, in \, the \, cell)}{(area \, of \, the \, region)/(total \, area \, of \, the \, cell)}$$

Here the area of a cell refers to the two-dimensional area of the cell from the differential interference contrast (DIC) image. The area of the nucleoid region was defined from the Hoechst image (nucleoid staining) and calculated by our custom MATLAB code (*Reyer et al., 2018*). Membrane region was determined as the boundary region from the DIC image, and the cytoplasm region was defined as the total cell region minus the nucleoid and the membrane regions.

## Population analysis

The analysis of mRNA-associated and mRNA-free population of Hfq was performed by double population fitting of the cumulative probability density function (CDF) of one-step squared displacement (*osd*$^2$) according to a previous report (*Bettridge et al., 2021*):

$$\text{CDF}\left(osd^2\right) = 1 - \sum_{i=1}^{n} P_i e^{-osd^2/4D'_i t}$$

where $n$ is the number of diffusion states, $D'_i$ is the diffusion coefficient of $i^{th}$ state, $P_i$ is the fraction of $i^{th}$ state population, and $\sum_{i=1}^{n} P_i = 1$. We found that a two-state model (n=2) fit better than one-state model (n=1), whereas a three-state model (n=3) did not further improve the fitting (*Figure 2—figure supplement 2a*). Therefore, we used a two-state model for fitting all Hfq tracking data, and the fast-diffusing state ($D_1$, $P_1$) was assigned as the mRNA-free fraction and the slow-diffusing state ($D_2$, $P_2$) assigned as the mRNA-associated. CDFs of rifampicin treatment cases in the WT *rne* background were fit first, the $D_1$ range of rifampicin treatment cases under WT *rne*, that is, $<D_{1,rif}> \pm 2 \times std\left(D_{1,rif}\right)$, was then used to constrain the $D_1$ values in the fitting for all other cases (*Figure 2—figure supplement 2c* and *Supplementary file 1*). All CDF fittings were conducted in OriginPro with Levenberg-Marquardt iteration algorithm. *Osd* speed was calculated as $osd/\Delta t_0$ ($\Delta t_0$ is the time interval between two consecutive frames, *i.e.*, 5.76 ms). For comparison with D values from the linear fitting of MSD, one-step diffusion coefficient $D_i \equiv 4D'_i$ values were reported in *Supplementary file 1* and *Figure 2—figure supplement 2d*.

# Acknowledgements

We thank CK Vanderpool and X Ma for sharing the plasmid containing mMaple3 template, E Perozo for sharing the plasmid containing Lac repressor and operator system, AA Kossiakoff for sharing the plasmid containing the synthetic antibody, Y Wang for sharing MATLAB codes for extracting tracking trajectories, and the MRSEC Shared User Facilities at the University of Chicago (NSF DMR-1420709) for providing 3D printed imaging chambers. We thank the Service de Purification de Protéines de l'Université de Sherbrooke (SPP) for Hfq purification and the Plateforme RNomique de l'Université de Sherbrooke for ddPCR. J Fei acknowledges the support by the Searle Scholars Program and NIH Director's New Innovator Award (1DP2GM128185-01). Work in E Massé Lab has been supported by an operating grant MOP69005 from the Canadian Institutes of Health Research (CIHR) and NIH Team Grant R01 GM092830-06A1.

# Additional information

## Funding

| Funder | Grant reference number | Author |
|---|---|---|
| National Institutes of Health | 1DP2GM128185-01 | Jingyi Fei |
| Searle Scholars Program | | Jingyi Fei |
| National Institutes of Health | R01 GM092830-06A1 | Eric Massé |
| Canadian Institutes of Health Research | MOP69005 | Eric Massé |

The funders had no role in study design, data collection and interpretation, or the decision to submit the work for publication.

## Author contributions

Seongjin Park, Karine Prévost, Marie-Claude Carrier, Data curation, Formal analysis, Validation, Visualization, Methodology, Writing - review and editing; Emily M Heideman, Data curation, Formal analysis, Validation, Visualization, Writing - review and editing; Muhammad S Azam, Data curation, Validation, Writing - review and editing; Matthew A Reyer, Software, Formal analysis, Validation, Visualization; Wei Liu, Formal analysis; Eric Massé, Resources, Data curation, Formal analysis, Supervision, Funding acquisition, Validation, Investigation, Visualization, Methodology, Project administration, Writing - review and editing; Jingyi Fei, Conceptualization, Resources, Data curation, Software,

Formal analysis, Supervision, Funding acquisition, Validation, Investigation, Visualization, Methodology, Writing - original draft, Project administration, Writing - review and editing

**Author ORCIDs**
Seongjin Park (iD) https://orcid.org/0000-0002-4298-1770
Marie-Claude Carrier (iD) https://orcid.org/0000-0002-3878-4141
Eric Massé (iD) https://orcid.org/0000-0002-5253-1415
Jingyi Fei (iD) https://orcid.org/0000-0002-9775-3820

**Decision letter and Author response**
Decision letter https://doi.org/10.7554/eLife.64207.sa1
Author response https://doi.org/10.7554/eLife.64207.sa2

## Additional files

### Supplementary files
- Source code 1. MATLAB scripts for tracking analysis and MSD/$osd^2$ calculation.
- Supplementary file 1. List of all tracking data sets used to extract mRNA-associated fractions in this study.
- Supplementary file 2. List of all strains and plasmids used in this study.
- Supplementary file 3. List of all oligonucleotides used in this study.
- Transparent reporting form

### Data availability
All the numeric data for each plot/graph and fitting results are provided in Supplementary file 1 or as source data. The MATLAB scripts for analysis are provided as source code.

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
