## [Decision Letter]

**Acceptance summary:**

The Hfq RNA chaperone protein modulates the stabilities and activities of base pairing small RNAs as well as mRNAs in many bacteria. Park et al. use single molecule tracking in combination with genetic manipulations and supporting biochemical experiments to study Hfq and its interactions with small RNAs, mRNAs and the RNase E ribonuclease in vivo, gaining further insights into the mechanism of Hfq-mediated regulation.

**Decision letter after peer review:**

[Editors’ note: the authors submitted for reconsideration following the decision after peer review. What follows is the decision letter after the first round of review.]

Thank you for submitting your work entitled "Dynamic interactions between the RNA chaperone Hfq, small regulatory RNAs and mRNAs in live bacterial cells" for consideration by *eLife*. Your article has been reviewed by a Senior Editor, a Reviewing Editor, and three reviewer. The following individual involved in review of your submission has agreed to reveal their identity: Jie Xiao (Reviewer #2).

Our decision has been reached after consultation between the reviewers. Based on these discussions and the individual reviews below, we regret to inform you that your work will not be considered further for publication in *eLife* at this point.

This study examining the diffusion of Hfq in vivo is important given the discrepant reports regarding Hfq localization and interactions with RNase E. However, the discrepant reports also are the reason it is so critical that the conclusions drawn in the current paper be strongly supported. All three of the reviewers raised significant but valid concerns. If you think you can address their comments (which will require additional experiments and analyses and is expected to take longer than two months), you could resubmit a substantially revised version of this study for consideration as a new manuscript at *eLife*.

Reviewer #1:

This manuscript provides data on the diffusion of Hfq in vivo under a number of conditions, using a tagged Hfq and super-resolution imaging. The authors compare the movement of Hfq with and without treatment with rifampicin, as well as in a number of Hfq mutants, and find differences in diffusion that they use to conclude that the protein is generally associated with mRNAs, with the distal face contributing most of the binding. in vitro and previous functional tests of sRNA degradation and regulation are in agreement with many of the findings here- that mRNAs mostly bind to the distal face of Hfq, that there is likely in vivo competition between some (class II) sRNAs and mRNA binding, and that RNase E contributes to degradation of the mRNAs. Work published by Elf and coworkers (Persson et al., 2013) and cited here showed faster diffusion after rifampicin treatment, as found here. In the previous work the authors suggested three states, the slowest moving due to Hfq interacting with mRNA during transcription, discussed briefly here; understanding to what extent the data presented here may be affected by co-translational regulation would be very useful. Other conclusions reached in the current work (role of Hfq in degradation of mRNAs, independent of sRNAs and suggestion of the presence of monomeric forms of Hfq binding mRNA) are more novel but will require further analysis to be fully convincing.

1) The control for the movement of tagged Hfq is the free mMaple3. Given that Hfq is a hexamer, and presumably thus carries 6 tags as well as the six copies of Hfq, the MW of these two proteins will be very different. It would be useful to have a more comparable control (another hexamer that doesn't interact with RNA?). This becomes particularly important in interpreting the effect of rifampicin, central to much of the paper; is it clear that the effect on Hfq movement is not reflecting a general change in the cellular milieu?

2) Subsection “Binding of mRNAs to Hfq decreases its diffusivity primarily through the distal face of Hf”: Is the interpretation in Figure 2B that with rifampicin, there is no mRNA-bound Hfq?

3) Subsection “Most Hfq proteins are occupied by mRNAs in the cell during exponential growth”: I did not understand the argument being made here about Hfq existing as a monomer. This is supposed to be only in the absence of RNA binding? Absence of mRNA binding? There is reason to believe, from previous rifampicin chase experiments, that sRNAs will still be bound to Hfq after 15' of rifampicin treatment. Do these still bind to this subset of monomers? Where did the number for a 2-4 fold change in MW come from? Why is that 50-100 kDa (of monomer MW? Hfq with single tag? Or from somewhere else?)? Is the same conclusion made for the distal site mutations of Hfq, in which (Figure 2D), apparently only one third are associated with mRNA. In this case, again, sRNAs should be continuing to bind to Hfq (in the absence of rifampicin). If this suggestion of monomers for Hfq existing in the cell is of importance/significance, it needs to be better explained.

4) Figure 3, Interactions with and role of RNase E: There are multiple issues here, and it is difficult to sort them out as presented.

a) The rne131 and rne14 mutant D values seem to be similar to those shown in Figure 2 for rne+ cells, although the direct comparison was not provided. Does that suggest that RNase E interactions are not affecting diffusion (is not part of a complex)? It is a large enzyme, supposed to associate with the membrane. This should be commented on.

b) In Figure 3b, the mRNA associated fraction of 40-50% after rifampicin treatment is noted; is there data or just an assumption that this value is extremely low in the rne+ cell? I could not find that data anywhere.

c) Why isn't there a comparison for the mRNA associated fraction (and D values) in the combination of Y25D and the rne mutants? This might help to determine if the effects of the rne mutants reflect more RNA in general or the extent of mRNA bound to Hfq.

d) Figure 3C: Please relabel; left panel grey bars is not a ratio but just the half-life of cells WT for hfq and rne, correct? This was not immediately obvious. The interpretation of the (very modestly) increased half-life in the Y25D cells effects on the half-lives in Figure 3C is interpreted as sRNA-independent since the well-studied sRNA regulators have been deleted. What is the result for Q8A (unable to bind most known sRNAs) vs. Q8A rne? Is the half-life similar to fully WT, as expected if this were truly sRNA independent? Is it totally clear that there are no other (3' UTR) sRNA regulators? Something further is necessary to interpret this result as the authors would like to.

e) In the model the authors are suggesting, should all of sodB or most of sodB, for instance, be bound to Hfq for RNAse-E dependent turnover?

5) Does displacement of mRNAs from Hfq (as with ChiX in Figure 4) lead to a change in the half-life of the mRNAs as the data in Figure 3 might predict?

6) Figure 5: Is it surprising that the levels of ChiX are the same in the Q8A mutant and in the cases where AANs are deleted? Does this reflect the very high level overexpression? Are the levels of the RNAs measured under conditions that at all parallel the imaging that is providing the diffusion data, in terms of growth and induction conditions and time? I found it difficult to determine this.

7) Discussion: This statement (that the authors directly observed the transition of major interaction partners in response to cellular changes) is misleading. The authors only looked at exponential growth (what is the evidence that there are fewer sRNAs expressed then?), and do not really have any evidence that sRNAs are not also present on Hfq – they just cannot detect them by their method. The one changing condition tested is rather drastic – with rifampicin. Are there differences seen under more physiological conditions?

8) Is it possible to extrapolate the data to determine how many Hfq hexamers are detectable, in particular whether all classes of Hfq (bound to mRNA or not) are being seen? Does the slow-diffusing, presumably mRNA or mRNA/ribosome bound Hfqs represent all Hfqs?

9) Discussion: This statement (that the observations here will provide possible tools for use in eukaryotes) has very little relationship to the work presented and is not supported by anything in this paper. I really don't think it belongs here.

Reviewer #2:

In this work, Park et al., characterize the interactions between RNA chaperone Hfq with both mRNAs and sRNAs using single molecule tracking under genetic and drug manipulations, which are supported by biochemical assays. The major findings are that Hfq can regulate the stability of some mRNAs through its interaction with RNase E, that class I sRNAs are able to co-occupy mRNA-associated Hfq molecules, and that class II sRNAs displace mRNAs from Hfq, which requires an AAN motif. The provides strong in vivo evidence of several previously proposed mechanisms of Hfq and sRNA-mediated post-transcriptional regulation. However, some analyses could be redone to better support the author's conclusions. In addition, some critical information is missing. I recommend the work for publication on the condition these concerns are addressed.

- The authors need to provide sufficient statistics for the reviewer to determine validity and significance. For example, how many trajectories were collected from how many cells for each condition? How many cells were imaged using FISH? How many trajectories and/or single step displacements were used for the fitting? These numbers need to be provided in the figure legends or supplementary table.

- The authors plot the osd distribution and use this to calculate the mRNA-associated fraction of Hfq, but use the overall MSD vs time to calculate the D of Hfq. The distribution of osd appeared to be lognormal sine the authors used Gaussian of log(osd) to do the fitting. If it is indeed lognormal, it means that most likely there are more than one Gaussian distributed population. The authors should use osd distribution (not log) or CDF to calculate Ds and subpopulations, and compare with that obtained from MSD, which is an average measurement. Such analysis should be conducted for all conditions.

- The authors should exert caution to use the power law distribution of diffusion coefficient and molecular weight in bacterial cells to interpret the association of Hfq-bound mRNA with ribosomes. The D for Hfq under NT condition is too small (~ 0.5 um2/s) compared to what would be expected form the molecular weight of Hfq-mRNA complex (~ 550 kD), therefore the authors interprets that a fraction of the associated mRNAs are translated by ribosomes. Additionally, because rif-treated Hfq showed higher diffusion coefficient compared to what should be expected from its MW, the authors concluded that Hfq could dissociate into monomers when not bound to RNAs. It is known that proteins with different surface charges and mRNA molecules diffuse differently in bacterial cells and also respond to different stress responses. They do not necessarily scale with the expected MW especially at high MW. See Kumar et al., 2010, Lampo et al., 2017, and Schavemaker et al., 2017. To substantiate the authors' claims, translation inhibitor and Hfq hexamer mutants should be used. Similarly, subsection “sRNAs can displace Hfq from mRNAs in a face-dependent manner” "sRNA-bound Hfq-mMaple3 has similar diffusivity as free Hfq-mMaple3" based on similar MW is not substantiated.

Reviewer #3:

This manuscript provides single-particle tracking data that describes the movement of the RNA-binding protein Hfq in living *E. coli* cells. The authors investigate how the diffusion of Hfq is affected by the presence of RNA, the ability of Hfq to interact with RNA, and mutations in the major endoribonuclease RNase E. The authors also investigate how overexpression of different small RNAs affects Hfq diffusion.

Although the tracking data is of high quality, this manuscript suffers from several problems. Most worrying is that the authors interpret their data in an often careless manner. The manuscript contains many instances of over-interpretations, and some interpretations of data that are entirely incorrect.

This reviewer is particularly concerned about the emphasis on the role for Hfq in mRNA destabilization via recruitment of Hfq. It seems this is the only new message in this paper and thus the most critical and important one. As detailed below, most aspects of this model hinge upon assumptions that are contradicted by available biochemical data from the Luisi lab, or have not been addressed by clear-cut unambiguous experiments in this manuscript.

1) The authors base many of their interpretations on the assumption that Hfq interacts directly with RNase E. From reading the manuscript, it appears as if a direct Hfq-RNase E interaction is a well-established fact. This is not correct. In fact, the interaction between Hfq and RNase E is highly controversial. According to work by the Aiba lab, Hfq and RNase E interact directly without a requirement for RNA (e.g. Ikeda et al., 2010). In contrast, the Luisi lab has shown that removing RNA from purified Hfq abolishes the Hfq-RNase E interaction (Bruce et al., 2008), and further showed by carefully conducted biochemical experiments that re-constituted Hfq-RNase E complexes only form when Hfq is pre-bound to an sRNA. According to Luisi's data, it is the sRNA (not Hfq) that directly interacts with RNase E (Worrall et al., 2018). Unfortunately, the authors have based a large part of their experiments and conclusions on this controversial assumption. Therefore, the authors need to explicitly state that it is unclear whether Hfq can bind RNase E directly, or whether this interaction is mediated through RNA. The data in this manuscript need to be re-interpreted in light of the uncertainty of the Hfq-RNase E interaction.

2) On the same topic, subsection “Hfq is deficient in releasing mRNAs without interactions with RNase E”: "Hfq has been demonstrated to interact with the C-terminal scaffold region of RNase". As mentioned above, it is unclear whether this interaction is direct or occurs via RNA. In fact, one of the papers (Bruce et al.,) cited after this sentence claims that the interaction is dependent on RNA, which is the complete opposite of the authors' statement.

3) The authors claim that rifampicin "inhibits transcription and results in the loss of most cellular mRNAs". A more correct wording would be that rifampicin treatment inhibits transcription and leads to a global loss of RNA. For instance, many Hfq-binding sRNAs have half-lives shorter than 15 minutes (e.g. Vogel et al., 2003). While the measured increased diffusion of Hfq after rifampicin treatment most likely indicates a higher fraction of Hfq free from RNA, it does not inform on the classes of RNA that contribute to this binding. Either the authors need to explicitly show that mRNAs are preferentially lost over other RNA classes after 15 minutes of rifampicin treatment, or the term "mRNA-associated fraction" used throughout the manuscript should be changed to "RNA-associated fraction".

4) The authors claim that mutations in the distal phase of Hfq lead to "a large increase in the diffusivity under NT condition". This is a clear overstatement; according to Figure 2C, the increase compared to WT is less than two-fold.

5) Subsection “Most Hfq proteins are occupied by mRNAs in the cell during exponential growth”: "Considering the average length of bacterial mRNAs to be 1 kb (~330 kDa), and Mw of bacterial ribosome (~2.5 MDa), this reduction in D supports the interpretation that a significant fraction of WT Hfq proteins are associated with mRNAs in the NT case, and that a fraction of the associated mRNAs are translated by the ribosomes." Do the authors imply that the majority of Hfq is associated with mRNA alone? An alternative explanation would be that most Hfq hexamers are simultaneously bound to mRNA and sRNA. The data does not inform on the nature of the complexes Hfq is involved in, other than that they contain RNA (according to the measured changes in diffusivity during rifampicin treatment). This should be clearly stated in the text.

6) Subsection “Hfq is deficient in releasing mRNAs without interactions with RNase E”: "The *rneΔ14* mutant has a smaller fraction of the C-terminal scaffold (residues 636-845) deleted, encompassing the Hfq binding region." The deleted part of RNase E also encompasses two RNA-binding domains, as well as binding sites for proteins RhlB and enolase. The differences in activity between WT and *rneΔ14* could thus stem from impairment of many different interactions and do not specifically report on the loss of a potential interaction with Hfq.

7) Subsection “Hfq is deficient in releasing mRNAs without interactions with RNase E”: "In both RNase E mutant backgrounds, the diffusivity of Hfq-mMaple3 became less sensitive to transcription inhibition by rifampicin compared to the WT rne case (Figure 3A and Figure 2C)." The authors should show the data for strains with WT and mutant RNase E in the same graph for easier comparison. In addition, a statistical analysis should be provided to test whether the claimed differences are significant.

8) Subsection “Hfq is deficient in releasing mRNAs without interactions with RNase E”: "40-50% of Hfq-mMaple3 remained mRNA associated upon rifampicin treatment in the RNase E mutant backgrounds (Figure 3B)." I could not find the corresponding numbers for the strain with WT RNase E. Please provide these numbers in the figure and/ or in this part of the text for clarity.

9) Subsection “Hfq is deficient in releasing mRNAs without interactions with RNase E”: "These observations suggest that without the Hfq-RNase E interaction, more mRNAs remain bound to Hfq, indicating that Hfq may help deliver the associated mRNA to RNase E for degradation." This is a very far-reaching interpretation of the data presented in Figure 3. There is no evidence for Hfq-mediated delivery of RNase E in Figure 3. I strongly advise the authors to use a more careful interpretation of the data.

10) In Figure 3C, the authors show half-life measurements of several mRNAs that are regulated by sRNAs. For these experiments, strains with corresponding sRNA gene deletions were used. I assume that the rationale for this was to avoid putative differences due to impaired sRNA regulation. This is not a stringent strategy, as it is unknown whether other sRNAs target these mRNAs. Moreover, from the bar charts in Figure 3C, it appears that mRNA half-lives increase when Hfq carries a distal face mutation. However, in the Northern blots used to create the bar charts (corresponding supplementary figure), there seem to be very small (if any) differences in half-lives between WT and mutant Hfq. For transparency, and easier interpretation for the reader, the authors should (instead of bar charts) plot the log10-transformed relative band intensities versus time, and show all data points (not only error bars). They should also include the corresponding Northern blots in Figure 3 along with the quantifications.

11) The authors propose that Hfq, through an interaction with RNase E, promotes mRNA degradation in an sRNA-independent fashion. They also suggest that "mRNA-occupied Hfq proteins are in standby mode for sRNA binding if needed". If this were correct, one would expect that a high cellular concentration of a Class II sRNA, that is a strong competitor for Hfq's interaction with mRNAs, would result not only in displacement of Hfq from many mRNAs, but also thereby increase their stability. ChiX is a strong competitor for distal phase binding that should cause such an effect. The authors interpret their data to imply that ChiX overexpression results in displacement of Hfq from many mRNAs. However, they do not provide evidence that this displacement results in general mRNA stabilization, which should be the outcome according to their model. In fact, induction of ChiX to a high intracellular concentration from a plasmid resulted in downregulation of one specific mRNA (ybfM), rather than a stabilizing effect on many mRNAs (Rasmussen et al., 2009). It is very surprising that the authors did not cite this paper.

12) Several reports have shown that the 5' moiety of RNA substrates largely influences on both, RNase E cleavage efficiency and specificity of cleavage site selection (e.g. Mackie, 1998, Jiang and Belasco, 2004, Bandyra et al., 2012); RNAs carrying a 5' monophosphate are substantially better substrates than those with a 5' tri-phosphate. Regarding the Hfq-dependent mRNA degradation proposed by the authors, do these mRNAs need to be decapped for RNase E to degrade them? Or does the degradation go through the substantially less efficient internal cleavage route? The authors should discuss both their data and their models with respect to what is known about the cleavage activity/ specificity of RNase E. They should also cite the most seminal papers on this subject.

13) Regarding the EMSA shown in Figure 4. Subsection “sRNAs can displace Hfq from mRNAs in a face-dependent man”: "Results show that RyhB cannot displace the radiolabeled ptsG from Hfq, but rather generates an additional upper-shifted band compared to the band of ptsG-Hfq complex, supporting that RyhB and ptsG can co-occupy Hfq". This is a very creative interpretation of an inconclusive result. What can be deduced from the gel picture is that the band representing the Hfq-ptsG complex becomes weaker at the two highest RyhB concentrations. The reason for this could be either that RyhB displaces ptsG, or that a ternary complex is formed. The design of the experiment makes it impossible to judge whether the former is happening; since the majority of ptsG mRNA is not in complex with Hfq (even in the absence of RyhB), it is impossible to judge whether high concentrations of RyhB results in increased free ptsG mRNA. Regarding the latter possibility (which is put forward by the authors), the "upper-shifted band" is barely visible and do not by any means reach the intensity of the ptsG-Hfq band, which would be the case if the major effect of RyhB addition would be the formation of a ternary complex.

14) Subsection “sRNAs can displace Hfq from mRNAs in a face-dependent man”: "In addition, droplet digital PCR (ddPCR) performed in the same conditions as the diffusivity assays showed that RyhB level was comparable to ChiX (Figure 4D)." This is not correct. According to Figure 4d ChiX is almost ten times more abundant than RyhB. The same incorrect statement is repeated later in this subsection.

15) Subsection “sRNAs can displace Hfq from mRNAs in a face-dependent man”: "EMSA and ddPCR results suggest that both in vitro and in vivo, RyhB can effectively access mRNA-occupied Hfq through co-occupying Hfq from the proximal face." This is an unsubstantiated and probably incorrect interpretation of the results. The EMSA does not provide evidence for a ternary Hfq-ptsG-RyhB complex. In what way does the ddPCR result inform on binding of RyhB to mRNA-bound Hfq?

16) Conceptually, there is no apparent reason why Hfq would not interact with RNA undergoing transcription. A previous Hfq tracking study in *E. coli* indeed reported a three-state model in which the slowest state was interpreted as Hfq bound to RNAs during transcription (Persson et al., 2013). The authors should comment on this finding with regard to their own results. Do the data presented in the current manuscript fit with the previous model? If not, why not?

[Editors’ note: further revisions were suggested prior to acceptance, as described below.]

Thank you for submitting your article "Dynamic interactions between the RNA chaperone Hfq, small regulatory RNAs and mRNAs in live bacterial cells" for consideration by *eLife*. Your article has been reviewed by Gisela Storz as a Reviewing and Senior Editor and three reviewers. The following individual involved in review of your submission has agreed to reveal their identity: Jie Xiao (Reviewer #2).

The reviewers have discussed the reviews with one another and the Reviewing Editor has drafted this decision to help you prepare a revised submission.

Summary:

In this resubmission, Park et al., use single molecule tracking in combination with genetic manipulations and supporting biochemical experiments to probe Hfq-mediated mRNA regulation in live *E. coli* cells. They added some critical control experiments, including investigating whether RppH (i.e. 5'-end decapping) is required for RNase E-Hfq mediated mRNA degradation, and whether ChiX overexpression can modulate mRNA turnover independently of Hfq-RNase E interaction. The connection between their single-molecule tracking results with actual underlying biology is strong. The authors also revised some of their analyses. Additionally, the interpretations of the data are much more careful, and alternative interpretations are discussed. The uncertainty regarding a direct interaction between Hfq and RNase E is clearly stated. The authors also acknowledge that the slow state of diffusion may represent not only Hfq-mRNA complexes but also Hfq-mRNA-sRNA or other complexes. Overall, these changes substantially improved the strength of the arguments made, and the study will contribute significantly to the understanding of Hfq-mediated regulation of mRNA.

Essential revisions:

However, two critical points still need to be addressed:

1) One of the major findings put forward by the authors, which is highlighted both in the abstract and in the schematic Figure 8, is the suggestion that binding of Hfq to mRNAs can recruit RNase E for (sRNA-independent) mRNA degradation. The authors provide live cell Hfq tracking data in strains with mutations in the Hfq distal and proximal phases, combined with full-length or mutant RNase E (Figure 4). The tracking data show that the differences in Hfq diffusion, which are observed between WT and mutant RNase E upon rif treatment, are abolished when Hfq has the distal phase mutation. These data are sound and convincing but do not directly address whether the observed effects are indeed due to mRNA degradation.

To directly monitor effects on mRNA degradation, the authors provide Northern blot data monitoring the half-life of three selected mRNAs (Figure 5). According to the authors, and many previous studies, the distal face mutation primarily impairs Hfq-mRNA interactions, while the proximal face mutant primarily impairs Hfq-sRNA interactions, at least when considering Class I sRNAs. In Figure 5, the presented data show that, in the strain expressing WT RNase E, both the distal and the proximal face mutants lead to increased mRNA half-lives. It is well established that sRNA-dependent (Class I sRNAs) regulation requires Hfq to contact both the sRNA and the mRNA. If Hfq-dependent degradation of mRNAs where to be mediated strictly through sRNAs, the increase half-life should be the same for the distal and proximal mutants (binding to both RNAs are required). However, according to the authors, the data presented in Figure 5 shows a greater increase in mRNA half-lives with the distal face mutant compared to the proximal face mutant, and is interpreted as representing Hfq-dependent and sRNA-independent degradation by RNase E. From this follows that the contribution of a sRNA-independent effect on mRNA half-lives can be deduced from the difference between the values obtained in the distal mutant and in the proximal mutant, while the difference between WT Hfq and the distal face mutant is the sum of the sRNA-dependent and sRNA-independent effects on mRNA degradation. In other words, if the authors' hypothesis is correct, there should be a significant difference in mRNA half-life between the distal and proximal face mutants, and this difference should be abolished if Hfq cannot interact with RNase E.

Unfortunately, the data provided in Figure 5 do not provide unequivocal evidence for the following reasons: (i) too few replicates: some of the mean values were calculated from only two data points, therefore the standard deviations are not meaningful (ii) there is no information on how many replicates where used for calculating each specific mean value, making it impossible to judge the how reliable specific mean values and standard deviations are, (iii) there is no statistical analysis provided to ensure that the differences are significant (which on the other hand is not possible with only two data points). For these reasons, it is impossible to judge whether the differences between Y25D and Q8A in the WT and rne131 mutant are significant. To test whether the hypothesis is correct, the authors need to provide at least three (four would be advisable) replicates for each mean value, and use statistical tests to assess whether the proposed differences are indeed significant. To increase the possibility for the reader to judge the data, each bar showing mean values in Figure 5d should to be overlaid with the value of each data point used for calculating the mean.

2) While the analysis is improved, there appeared to be a misunderstanding on how to use CDF to fit displacement to extract the diffusion coefficients and population percentages. The authors used the CDF fitting of the apparent speed of one step displacement (osd) to extract the average speed and population percentage of Hfq. The authors cited the Yang, 2019 paper as the source. Do note that in Yang, 2019, the authors were measuring the directional moving speed but not random diffusion. For non-processively diffusing molecules, the diffusion coefficient D should be used. While the speed CDF from osd can also be used to extract different populations based on the difference in speed, it is different from classifying molecules based on their apparent diffusion coefficients (note that only osd2, but not osd, is proportional to D). Two molecules diffusing at different speeds (with random orientations) will have different Ds, but the quantitative difference between the two Ds is not the same as that between the two speeds (again, osd2 , but not osd, is proportional to D). Therefore, the classification of fast and slow diffusing molecules based on diffusion coefficient could be different from that based on speed. Furthermore, the mean squared displacement (MSD) measures how far the molecule diffuses away over different time lags. MSD is also an averaged measurement of molecules of different Ds, and not particularly accurate when there are at least two populations of different Ds, which is the case the authors are trying to establish. See https://link.springer.com/protocol/10.1007/978-1-59745-513-8_14 for a review, and Bettridge, et al., https://doi.org/10.1111/mmi.14572, in the supplemental notes for a practical guide. The fractionation of subpopulations based on diffusion coefficient would be important for the authors to make the argument of whether mRNA binding would lower the D or vice. versa. If the difference in speed is fairly large, I suspect that the major conclusions should still hold if the authors switch to the analysis of CDF of osd2, but the correct analysis should be provided.

---

## [Author Response]

[Editors’ note: the authors resubmitted a revised version of the paper for consideration. What follows is the authors’ response to the first round of review.]

We would like to thank all three reviewers for their careful consideration and constructive suggestions, particularly under the difficult situation of COVID-19 outbreak. We have now significantly revised the manuscript based on the reviewers’ suggestions, including new data and analysis. As some of the major concerns are shared by more than one reviewer, we would like to summarize the corresponding changes first before providing point-to-point responses on specific critiques.

Essential revisions:

(1) Data analysis for determining mRNA-associated and mRNA-free fraction of Hfq

Several critiques raised by reviewers 1 and 3 regarding the mRNA-associated and mRNAfree fraction of Hfq under various conditions are due to our fault of not presenting the analysis in a clear enough fashion. In our previous analysis, we assumed under rifampicin (Rif) treated cases, Hfq proteins were all free of mRNAs, and fit the one-step displacement (*osd*) histogram for Rif treated cases with single Gaussian. Then with the mean *osd* value for the Rif treated case, we fit the no treatment (NT) case with two Gaussians to estimate the mean *osd* value for the mRNA-bound Hfq, and the associated fraction mRNA-free and mRNA-associated Hfq. We thank Dr. Jie Xiao (reviewer 2) for her suggestion on alternative analysis. We have accordingly applied the fitting of cumulative probability density function (CDF) of the one-step directional (osd) speed to recalculate the mRNA-associated and mRNA-free fraction of Hfq, as described in (Yang et al., https://doi.org/10.1101/850073). differences in the analysis are summarized as below:

(1) Data are all described well with two-population models, better than single-population fitting. Comparison is presented in Figure 2—figure supplement 2.

(2) When applying the new analysis method, we made one additional change in building trajectories from the raw imaging: changing the maximum one-step distance from the original cutoff of 250 nm to 400nm. We made this change, because we realized our initial distance cutoff of 250 nm was too stringent, which caused significant loss of fast-diffusing population, particularly for Rif treatment cases. We therefore increased the cutoff to 400 nm. As presented in Figure 1—figure supplement 3, for NT and Rif cases, particularly Rif case, changing the cutoff distance from 250 nm to 400 nm included significantly more fast-diffusing population (the right four plots in panel b). As a control, we compared the two distance cutoffs for fixed cells, and the difference was minimal (the leftmost two plots).

(3) The analysis and fitting are presented in Figure 2—figure supplement 2 and Supplement file 1. Briefly, CDFs of all Rif cases (for WT Hfq and all Hfq mutants) were fitted with double lognormal populations, and we obtained the relative populations (probability values) of the fast-diffusing state and the slow-diffusing states, which we assigned as mRNA-free and mRNA-bound Hfq populations, respectively. Rif cases for all mutants gave consistent fitting results. To better estimate the percentage of the two populations for the NT cases, we constrained the fitting parameters for the fast population during fitting. Importantly, the population average of the diffusion coefficient from the new analysis is consistent with the ensemble diffusion coefficient from linear fitting of MSD vs Δt curve in all cases (panel d in Figure 2—figure supplement 2).

All conclusions remain the same with the new analysis, with mostly minor changes in the actual numbers (diffusion coefficient and population percentage). One difference worth noticing is the mRNA-associated fraction (slow population) in Rif case, which on average is around 40%. mRNA-associated fraction in the Rif case is likely to be overestimated, or mRNA-free fraction in the Rif case is likely to be underestimated, because even with 400 nm distance cutoff, we may still likely miss some fast-diffusing population (shown in the figure above). However, we decided not to further increase the distance cutoff in the analysis, in order to prevent creating artificial trajectories. Nevertheless, this does not affect our conclusions, since all the trends of Hfq diffusion changes remained the same with the new fitting method.

(4) Related to the population analysis, reviewers 1 and 3 also brought up the analysis in a previous study on Hfq, in which an HMM-based method (named vbSTP) was used to estimate different diffusion states. Particularly, in addition to mRNA-free Hfq, and mRNAbound Hfq, vbSTP also revealed a third state representing Hfq associated with nascent mRNA during transcription. Indeed, when we applied vbSPT to our tracking data, we actually got very consistent results (Author response image 1).

In this figure, vbSPT plots for WT Hfq, NT case (left) and Rif case (right) are presented. Upper plots are from vbSPT fitting (software package is from the Elf’s lab) to our datasets, and the lower plots are from the corresponding paper (Persson et al., 2013, Figure 4). Please note that the software output as well as the manuscript by Persson et al., use the factor of 4 for MSD fitting, i.e., MSD=4*D*Δt. Thus, to compare with the diffusion coefficients in the above plots, our diffusion coefficients presented in the revised manuscript should be divided by 4. For example, in “WT Hfq, Rif case, we report 9±0.8 µm^2^/s and 0.9±0.4 for the faster and slower diffusion coefficients from CDF fitting (Figure 2—source data 1), which translate to 2.3 µm^2^/s and 0.23 µm^2^/s with the consideration of the factor of 4. These numbers are very comparable to our vbSPT outcomes, 2.1 µm^2^/s and 0.304 µm^2^/s Author response 1, top right). In the same way, we compare all these numbers in Author response image 2.

**Author response image 2. respfig2:** 

(Since we only fit two states for the CDF fitting method, there are only D_1_ and D_3_ values for NT case in our fitting). Overall, the manuscript by Persson et al., generally gave slightly faster diffusion coefficients, but these numbers are comparable for each case. Please note that these two studies used two different fluorescent protein taggings (mMaple3 for ours, and Dendra2 for Persson et al.,), possibly contributing to the differences of vbSPT outputs.While we got roughly consistent results using vbSPT, we had concerns on using this method to estimate the fraction of different population due to two reasons:

First, we found vbSPT may overfit the data sometimes, partially because the algorithm does not consider the localization uncertainty in imaging. For example, when we analyzed more than 10000 trajectories (well above the recommended minimal number of trajectories, 3000), from the NT case, vbSPT generates 4 states. The diffusion coefficient of the slowest state corresponds to a one-step displacement of 20nm, actually below our localization uncertainty (Author response image 3) (0.075 μm^2^/s of D value corresponds to (0.075x0.00576)/0.00576=3.6µm/s of diffusion speed, so between two imaging frames, this corresponds to 3.6 x 0.00576 = 0.*02* µm = 20 nm of displacement).

**Author response image 3. respfig3:** 

Similarly, Hfq-mMaple in fixed cells was fit by vbSPT into four states. Again, the one-step displacement in the slowest state is beyond our imaging accuracy.

**Author response image 4. respfig4:** 

Second, vbSPT does not allow users to constrain fitting parameters. In both their and our data, vbSPT tends to fit a smaller diffusion coefficient for the mRNA-free population in the NT case compared to Rif case (D3 value in NT vs D2 value in Rif), which would lead to an overestimation of the mRNA-free population in the NT cases. Based on these reasons, we prefer to use the analysis method suggested by reviewer 2.

(2) Reviewer 1 and 3 raised several concerns on the data and interpretations that lead to the conclusion that Hfq can regulate mRNA turnover in a sRNA-independent way. We thank both reviewers for their suggestion of several new experiments. We are summarizing the new experiments below:

(1) We added new imaging experiments on Y25D and Q8A Hfq mutants in the *rne* mutant backgrounds. The data are presented in the new Figure 4. The description of the results is as below:

“Hfq Y25D-mMaple3, as it is deficient in mRNA binding, showed minimal sensitivity to rifampicin treatment in the *rne* mutant backgrounds, the same as in the WT *rne* background (Figure 4A and B). These observations suggest that without the Hfq-RNase E interaction, more mRNAs remained bound to Hfq, and hint that Hfq-RNase E interaction may help recycle Hfq from the mRNA-associated form through degradation of mRNAs. To investigate whether the reduction of mRNA releasing from Hfq in the *rne* mutant background is primarily contributed by sRNAs, we imaged Hfq Q8A-mMaple3 in the *rne* mutant backgrounds. Q8A mutation of Hfq broadly disrupts the binding and stabilization of sRNAs^27^. If sRNA is the dominating factor in driving the mRNA releasing from Hfq, we would expect a significant increase in the mRNA-associated fraction of the Q8A mutant in the *rne* mutant backgrounds compared to the WT Hfq with rifampicin treatment. However, we did not observe any significant difference in the mRNAassociated fraction between Q8A mutant and the WT Hfq (Figure 4B), suggesting that Hfq-RNase E interaction can also drive the release of mRNA from Hfq independently of sRNAs.”

”

2) We also added new mRNA half-life measurements in the background of Hfq Q8A in addition to knockout of the relevant sRNAs. The data are presented in new Figure 5. The description of the results in the revised manuscript is as blow:

“To further exclude the contributions by potentially unknown sRNAs, we compared the lifetime of *sodB*, *ompX* and *sdhC* in the *hfq-mMaple3* Q8A background in addition to knocking out corresponding sRNAs (Figure 5). The half-lives of these mRNAs increased by 19%-46% in the *hfq* Q8A background compared to WT *hfq* background, smaller than the increase observed in the *hfq* Y25D background (Figure 5D). The increase of mRNA half-life in the *hfq* Q8A background can either be due to contributions by unknown sRNA regulators, or due to other possible regulatory pathways by Hfq through binding at the proximal face. One of such regulatory pathways may be Hfq-mediated polyadenylation, which involves binding of Hfq at the Rho-independent termination site and promotes mRNA degradation^9,10^. Despite these two possibilities, the increase in the mRNA half-life due to Y25D mutation cannot be fully explained by sRNA-mediated regulation. These results collectively support that besides the sRNA mediated pathway, Hfq can facilitate the turnover of certain mRNAs by binding to the mRNAs through the distal face and bridging them to RNase E for degradation.”

Based on the new results, we think our original interpretation still holds, that Hfq can regulate mRNA turnover in a small-RNA independent manner in addition to sRNA-mediated pathway. However, we make it more specific that this regulation is mediated through binding of mRNA at the distal face of Hfq.

Reviewer #1:This manuscript provides data on the diffusion of Hfq in vivo under a number of conditions, using a tagged Hfq and super-resolution imaging. The authors compare the movement of Hfq with and without treatment with rifampicin, as well as in a number of Hfq mutants, and find differences in diffusion that they use to conclude that the protein is generally associated with mRNAs, with the distal face contributing most of the binding. in vitro and previous functional tests of sRNA degradation and regulation are in agreement with many of the findings here- that mRNAs mostly bind to the distal face of Hfq, that there is likely in vivo competition between some (class II) sRNAs and mRNA binding, and that RNase E contributes to degradation of the mRNAs. Work published by Elf and coworkers (Persson et al., 2013) and cited here showed faster diffusion after rifampicin treatment, as found here. In the previous work the authors suggested three states, the slowest moving due to Hfq interacting with mRNA during transcription, discussed briefly here; understanding to what extent the data presented here may be affected by co-translational regulation would be very useful. Other conclusions reached in the current work (role of Hfq in degradation of mRNAs, independent of sRNAs and suggestion of the presence of monomeric forms of Hfq binding mRNA) are more novel but will require further analysis to be fully convincing.

We thank Reviewer 1 for her/his overall positive comments and useful feedback. As this reviewer mentioned above, the finding that Hfq might contribute to the regulation of mRNA turnover in the absence of the corresponding sRNA regulators is novel. In addition, our results provide strong *in vivo* support for some of the previous *in vitro* biochemical and functional tests and conclusions on Hfq-mRNA and Hfq-sRNA interactions. Finally, we also found that while class II sRNAs can directly compete with mRNA for binding to Hfq, class I sRNAs do not necessarily need to displace the mRNA, but can co-occupy with mRNA on Hfq through different binding sites. These results provide more concrete mechanisms to the previously suggested RNA exchange model.

1) The control for the movement of tagged Hfq is the free mMaple3. Given that Hfq is a hexamer, and presumably thus carries 6 tags as well as the six copies of Hfq, the MW of these two proteins will be very different. It would be useful to have a more comparable control (another hexamer that doesn't interact with RNA?). This becomes particularly important in interpreting the effect of rifampicin, central to much of the paper; is it clear that the effect on Hfq movement is not reflecting a general change in the cellular milieu?

We thank the reviewer for the good suggestion. To my knowledge, I have not seen similar control being used in any studies using single-particle tracking. We reasoned that to make it a solid control, it is better to find a protein that does not interact with DNA (since rifampicin also causes changes in DNA morphology) or RNA, nor naturally exists in *E. coli*. However, after discussing with colleagues with expertise in protein purification, we found it difficult to express a recombinant protein that meets these criteria, and is either hexamer or with a size of 200 kDa. Nevertheless, we generated two more controls with a total size of 80 kDa each, one being the scFv protein used in the SunTag technique, the other one being a synthetic antibody sAB-70. Both of them behave the same as the mMaple3 only control, i.e. do not show a difference in the diffusivity upon rifampicin treatment. The new data are included in Figure 2—figure supplement 1. We hope reviewer 1 can find these new controls satisfying. In fact, it is unlikely that the observed difference in Hfq-mMaple3 is due to change in the cellular milieu, as the difference we observed upon rifampicin treatment varies between different Hfq mutants. For the Y25D mutant, it is also most completely insensitive to rifampicin treatment, arguing against the possibility that the difference we observed in WT Hfq and other Hfq mutants are due to a general change in the cellular milieu.

2) Subsection “Binding of mRNAs to Hfq decreases its diffusivity primarily through the distal face of Hf”: Is the interpretation in Figure 2B that with rifampicin, there is no mRNA-bound Hfq?

We apologize for not being clear in the previous manuscript. In the previous analysis, we assumed that with high concentration of rifampicin treatment for 15 minutes, the cells will be depleted with mRNAs and therefore mRNA-bound Hfq is close to 0%. This assumption helped us on the fitting of the mRNA-bound fractions in the no treatment case. We have now reanalyzed all the data as suggested by reviewer 2, and the mRNA-bound fractions are reported now for all conditions in the new Figure 2 and Figure 4. The details on the new data analysis method are described in the new Figure 1—figure supplement 3 and Figure 2—figure supplement 2, and summarized in the Essential revision part 1 above.

3) Subsection “Most Hfq proteins are occupied by mRNAs in the cell during exponential growth”: I did not understand the argument being made here about Hfq existing as a monomer. This is supposed to be only in the absence of RNA binding? Absence of mRNA binding? There is reason to believe, from previous rifampicin chase experiments, that sRNAs will still be bound to Hfq after 15' of rifampicin treatment. Do these still bind to this subset of monomers? Where did the number for a 2-4 fold change in MW come from? Why is that 50-100 kDa (of monomer MW? Hfq with single tag? Or from somewhere else?)? Is the same conclusion made for the distal site mutations of Hfq, in which (Figure 2D), apparently only one third are associated with mRNA. In this case, again, sRNAs should be continuing to bind to Hfq (in the absence of rifampicin). If this suggestion of monomers for Hfq existing in the cell is of importance/significance, it needs to be better explained.

We apologize for the confusion. Our previous interpretation was based on the power-law relationship between molecular weight (M_w_) and the diffusion coefficient (D) (D = aM_w_^x^, where x = -0.33) that was experimentally observed in bacterial cells (reference 42). Comparing the diffusion coefficients of Hfq-mMaple3 and free mMaple, and the molecular weight of free mMaple3, we estimated that the molecular weight of Hfq-mMaple3 is (50-100kDa) based on the power-law relationship, which is smaller than Hfq-mMaple3 hexamer. Therefore, we concluded that part of the Hfq-mMaple3 may exist in monomer form. However, as reviewer 2 suggested, inference of M_w_ just based on D may not be robust, since diffusion coefficient is also dependent on other factors, such as surface charge of the protein. In the revised manuscript, we have removed the part of the Discussion, as it does not contribute to the major conclusions of our manuscript.

4) Figure 3, Interactions with and role of RNase E: There are multiple issues here, and it is difficult to sort them out as presented.a) The rne131 and rne14 mutant D values seem to be similar to those shown in Figure 2 for rne+ cells, although the direct comparison was not provided. Does that suggest that RNase E interactions are not affecting diffusion (is not part of a complex)? It is a large enzyme, supposed to associate with the membrane. This should be commented on.

We thank the reviewer for bringing up this point. We improved the analysis on the one-step diffusion speed as a function of cell coordinates (Figure 1B-D), in which we further separated the cell into nucleoid, cytoplasm and membrane region. Indeed, Hfq diffusion speed in the membrane region is slightly lower compared to nucleoid and cytoplasm, which is likely due to the interactions with RNase E localized in the inner membrane. However, since the population of Hfq diffusing into the inner membrane region is not significant compared to the total population of Hfq, the minor slowing down of diffusion speed is overweighed by Hfq population in nucleoid and cytoplasm. We cannot robustly resolve the populations of Hfq interacting with RNase E in the presence and absence of mRNAs. For visualization of the data, we provided a direct comparison of D values under WT and mutant RNase E in the new Figure 4.

b) In Figure 3B, the mRNA associated fraction of 40-50% after rifampicin treatment is noted; is there data or just an assumption that this value is extremely low in the rne+ cell? I could not find that data anywhere.

This is the same issue as the point #2 brought up by the reviewer -- we assumed that the mRNA associated fraction is close to zero after rifampicin treatment in the WT *rne* background. With this assumption, fitting of the rifampicin treated case in the *rne131* background generated 4050% of mRNA-associated fraction. We have now reanalyzed all the data based on reviewer 2’s suggestion. The mRNA-bound fractions are reported now for all conditions in the new Figure 2 and Figure 4. The details on the new data analysis method are described in the new Figure 1—figure supplement 3 and Figure 2—figure supplement 2, and summarized in the Essential revisions part 1 above.

c) Why isn't there a comparison for the mRNA associated fraction (and D values) in the combination of Y25D and the rne mutants? This might help to determine if the effects of the rne mutants reflect more RNA in general or the extent of mRNA bound to Hfq.

We thank the reviewer for the good suggestion. We have now added data in the background of Y25D/*rne*131 background (new Figure 4). Without rifampicin treatment, Hfq Y25D-mMaple3 in the *rne* mutant background demonstrated low mRNA-associated fraction, the same as the no treatment case of the Y25D in theWT *rne* background. With rifampicin treatment, Y25D in the *rne* background also showed the same efficiency in mRNA releasing as in the WT *rne* background, suggesting that this Y25D mutation does not contribute to the regulation of mRNA turnover by RNase E, presumably due to its deficiency in mRNA binding.

d) Figure 3C: Please relabel; left panel grey bars is not a ratio but just the half-life of cells WT for hfq and rne, correct? This was not immediately obvious. The interpretation of the (very modestly) increased half-life in the Y25D cells effects on the half-lives in Figure 3C is interpreted as sRNA-independent since the well-studied sRNA regulators have been deleted. What is the result for Q8A (unable to bind most known sRNAs) vs. Q8A rne? Is the half-life similar to fully WT, as expected if this were truly sRNA independent? Is it totally clear that there are no other (3' UTR) sRNA regulators? Something further is necessary to interpret this result as the authors would like to.

We have added the new half-life measurement using Q8A mutant in addition to small RNA knockout as the reviewer suggested. The new data are presented in new Figure 5, and the description is as below:

“To further exclude the contributions by potentially unknown sRNAs, we compared the lifetime of *sodB*, *ompX* and *sdhC* in the *hfq-mMaple3* Q8A background in addition to knocking out corresponding sRNAs (Figure 5). The half-lives of these mRNAs increased by 19%-46% in the *hfq* Q8A background compared to WT *hfq* background, smaller than the increase observed in the *hfq* Y25D background (Figure 5D). The increase of mRNA half-life in the *hfq* Q8A background can either be due to contributions by unknown sRNA regulators, or due to other possible regulatory pathways by Hfq through binding at the proximal face. One of such regulatory pathways may be Hfq-mediated polyadenylation, which involves binding of Hfq at the Rho-independent termination site and promotes mRNA degradation^9,10^. Despite these two possibilities, the increase in the mRNA half-life due to Y25D mutation cannot be fully explained by sRNA-mediated regulation. These results collectively support that besides the sRNAmediated pathway, Hfq can facilitate the turnover of certain mRNAs by binding to the mRNAs through the distal face and bridging them to RNase E for degradation.”

e) In the model the authors are suggesting, should all of sodB or most of sodB, for instance, be bound to Hfq for RNAse-E dependent turnover?

Our measurement cannot determine the percentage of individual mRNA species that are associated with Hfq, i.e., we do not know the percentage of *sodB* mRNAs that bind to Hfq. In fact, since many mRNAs compete for Hfq, it is actually unlikely that *sodB* mRNAs are close to being completely bound by Hfq. In this sense, our model only suggests that Hfq-RNase E-dependent degradation “contributes” to the endogenous turnover of some mRNAs, but is not the sole pathway for these mRNAs. Other RNase-E mediated degradation pathways clearly also contribute. We have emphasized this point in the second before the last paragraph of the Discussion.

5) Does displacement of mRNAs from Hfq (as with ChiX in Figure 4) lead to a change in the half-life of the mRNAs as the data in Figure 3 might predict?

We thank the reviewer for the good suggestion. We added new lifetime measurements in the presence of WT ChiX and ChiX with two AAN motif deleted, and found that the presence of WT ChiX can indeed increase the *sdhC* lifetime, similar to Y25D Hfq mutant, whereas the mutant ChiX that has reduced Hfq binding ability is not able to increase the lifetime as significantly as the WT ChiX. The new data is presented as new Figure 6.

“As our model suggests that binding of Hfq to the mRNA through the distal face can regulate the mRNA turnover, we reasoned that sRNAs that can effectively compete for Hfq binding against mRNAs may decoy Hfq from this regulatory function. To test this, we again used *sdhC* as an example, and measured its half-life in the presence of ChiX, which is a strong competitor for Hfq binding (Figure 3). In the presence of vector control, *sdhC* exhibited comparable half-life compared to the case without any plasmid (Figure 5D, Figure 6C and D). The presence of WT ChiX increased the half-life by ~70%, whereas the mutant ChiX without two AAN motif deleted only increased the half-life by ~42%, consistent with its reduced binding ability to Hfq (Figures 3a, 3b, 6c and 6d). These results further support our model of Hfq-mediated regulation of mRNA turnover, and demonstrate that the presence of strong Hfq binding sRNAs can modulate the strength of Hfq’s regulation.”

6) Figure 5: Is it surprising that the levels of ChiX are the same in the Q8A mutant and in the cases where AANs are deleted? Does this reflect the very high level overexpression? Are the levels of the RNAs measured under conditions that at all parallel the imaging that is providing the diffusion data, in terms of growth and induction conditions and time? I found it difficult to determine this.

The cell growth media and inducer concentration are the same for ddPCR and live-cell imaging as described in the Materials and methods. Minor difference is that cells were imaged after 45 min induction, whereas the total RNA was extracted after 1 hour induction. The reason for this is that imaging the live-cell samples took 10-15 minutes. Therefore, the actual induction times were comparable between these two experiments. Both ddPCR and FISH experiments confirmed that the relative levels of WT and mutant ChiX were very comparable (Figure 3D and Figure 3—figure supplement 2), perhaps due to plasmid-expressing levels being higher than endogenous abundance.

7) Discussion: This statement (that the authors directly observed the transition of major interaction partners in response to cellular changes) is misleading. The authors only looked at exponential growth (what is the evidence that there are fewer sRNAs expressed then?), and do not really have any evidence that sRNAs are not also present on Hfq – they just cannot detect them by their method. The one changing condition tested is rather drastic – with rifampicin. Are there differences seen under more physiological conditions?

“Cellular changes” also include the cases where we expressed specific sRNAs, and we observed that the sRNAs can compete for Hfq binding against mRNAs. We have removed that sentence from Discussion.

8) Is it possible to extrapolate the data to determine how many Hfq hexamers are detectable, in particular whether all classes of Hfq (bound to mRNA or not) are being seen? Does the slow-diffusing, presumably mRNA or mRNA/ribosome bound Hfqs represent all Hfqs?

Unfortunately, we are not able to determine the absolute copy number of Hfq, as the photoconversion of the fluorescent protein is not 100% during the time window of our imaging experiments. Since the changes we introduced, such as rifampicin treatment and sRNA expression, more specifically affect the association of Hfq with the mRNAs, we interpret that the shift from the slow population to fast population is due to removal of the mRNA from Hfq and the slow-diffusing population represent the mRNA-bound Hfq. It is possible that some fraction of the slow-diffusing population also includes other Hfq-involved complexes, such as DNA associated Hfq. But one needs to introduce mutation specific affecting Hfq-DNA interaction to extrapolate that population, which is beyond the scope of the current manuscript.

9) Discussion: This statement (that the observations here will provide possible tools for use in eukaryotes) has very little relationship to the work presented and is not supported by anything in this paper. I really don't think it belongs here.

We have deleted this part.

Reviewer #2:In this work, Park et al., characterize the interactions between RNA chaperone Hfq with both mRNAs and sRNAs using single molecule tracking under genetic and drug manipulations, which are supported by biochemical assays. The major findings are that Hfq can regulate the stability of some mRNAs through its interaction with RNase E, that class I sRNAs are able to co-occupy mRNA-associated Hfq molecules, and that class II sRNAs displace mRNAs from Hfq, which requires an AAN motif. The provides strong in vivo evidence of several previously proposed mechanisms of Hfq and sRNA-mediated post-transcriptional regulation. However, some analyses could be redone to better support the author's conclusions. In addition, some critical information is missing. I recommend the work for publication on the condition these concerns are addressed.

We thank reviewer 2 for her positive comments and useful feedback, particularly on the data analysis. We have changed our analysis based on her suggestion, described in Part 1 of Essential revisions above, and also addressed all her specific points in the manuscript.

- The authors need to provide sufficient statistics for the reviewer to determine validity and significance. For example, how many trajectories were collected from how many cells for each condition? How many cells were imaged using FISH? How many trajectories and/or single step displacements were used for the fitting? These numbers need to be provided in the figure legends or supplementary table.

We thank the reviewer for pointing out the issues. We have now added all the information to the source data and figure captions.

- The authors plot the osd distribution and use this to calculate the mRNA-associated fraction of Hfq, but use the overall MSD vs time to calculate the D of Hfq. The distribution of osd appeared to be lognormal sine the authors used Gaussian of log(osd) to do the fitting. If it is indeed lognormal, it means that most likely there are more than one Gaussian distributed population. The authors should use osd distribution (not log) or CDF to calculate Ds and subpopulations, and compare with that obtained from MSD, which is an average measurement. Such analysis should be conducted for all conditions.

We thank the reviewer for the good suggestion. We have reanalyzed all the data using the suggested method. We summarize the results from the new analysis in Essential revisions point 1, to make it easier to read for all three reviewers.

- The authors should exert caution to use the power law distribution of diffusion coefficient and molecular weight in bacterial cells to interpret the association of Hfq-bound mRNA with ribosomes. The D for Hfq under NT condition is too small (~ 0.5 um2/s) compared to what would be expected form the molecular weight of Hfq-mRNA complex (~ 550 kD), therefore the authors interprets that a fraction of the associated mRNAs are translated by ribosomes. Additionally, because rif-treated Hfq showed higher diffusion coefficient compared to what should be expected from its MW, the authors concluded that Hfq could dissociate into monomers when not bound to RNAs. It is known that proteins with different surface charges and mRNA molecules diffuse differently in bacterial cells and also respond to different stress responses. They do not necessarily scale with the expected MW especially at high MW. See Kumar et al., 2010, Lampo et al., 2017, and Schavemaker et al., 2017. To substantiate the authors' claims, translation inhibitor and Hfq hexamer mutants should be used. Similarly, subsection “sRNAs can displace Hfq from mRNAs in a face-dependent manner” "sRNA-bound Hfq-mMaple3 has similar diffusivity as free Hfq-mMaple3" based on similar MW is not substantiated.

We thank the reviewer for pointing this out. We agree that the interpretation on Mw based on the diffusion coefficient with the power-law relationship is not robust. Since the issue is brought up by all three reviewers, we removed this discussion from the manuscript, which does not affect any major conclusions of the manuscript.

Reviewer #3:This manuscript provides single-particle tracking data that describes the movement of the RNA-binding protein Hfq in living *E. coli* cells. The authors investigate how the diffusion of Hfq is affected by the presence of RNA, the ability of Hfq to interact with RNA, and mutations in the major endoribonuclease RNase E. The authors also investigate how overexpression of different small RNAs affects Hfq diffusion.Although the tracking data is of high quality, this manuscript suffers from several problems. Most worrying is that the authors interpret their data in an often careless manner. The manuscript contains many instances of over-interpretations, and some interpretations of data that are entirely incorrect.This reviewer is particularly concerned about the emphasis on the role for Hfq in mRNA destabilization via recruitment of Hfq. It seems this is the only new message in this paper and thus the most critical and important one. As detailed below, most aspects of this model hinge upon assumptions that are contradicted by available biochemical data from the Luisi lab, or have not been addressed by clear-cut unambiguous experiments in this manuscript.

We thank this reviewer for his/her critiques and constructive suggestions. We have addressed the comments and concerns from the reviewer as shown below, and significantly revised the manuscript. However, we would still like to emphasize the novelty and significance of our study here. In addition to the new finding that Hfq might contribute to the regulation of mRNA turnover in the absence of the corresponding sRNA regulators, as the reviewer mentioned, our results provide strong in vivo support for some of the previous in vitro biochemical and functional tests and conclusions on Hfq-mRNA and Hfq-sRNA interactions. Finally, we also found that while class II sRNAs can directly compete with mRNA for binding to Hfq, class I sRNAs do not necessarily need to displace the mRNA, but can co-occupy with mRNA on Hfq through different binding sites. These results provide more concrete mechanisms to the previously suggested RNA exchange model.

1) The authors base many of their interpretations on the assumption that Hfq interacts directly with RNase E. From reading the manuscript, it appears as if a direct Hfq-RNase E interaction is a well-established fact. This is not correct. In fact, the interaction between Hfq and RNase E is highly controversial. According to work by the Aiba lab, Hfq and RNase E interact directly without a requirement for RNA (e.g. Ikeda et al., 2010). In contrast, the Luisi lab has shown that removing RNA from purified Hfq abolishes the Hfq-RNase E interaction (Bruce et al., 2008), and further showed by carefully conducted biochemical experiments that re-constituted Hfq-RNase E complexes only form when Hfq is pre-bound to an sRNA. According to Luisi's data, it is the sRNA (not Hfq) that directly interacts with RNase E (Worrall et al., 2018). Unfortunately, the authors have based a large part of their experiments and conclusions on this controversial assumption. Therefore, the authors need to explicitly state that it is unclear whether Hfq can bind RNase E directly, or whether this interaction is mediated through RNA. The data in this manuscript need to be re-interpreted in light of the uncertainty of the Hfq-RNase E interaction.2) On the same topic, subsection “Hfq is deficient in releasing mRNAs without interactions with RNase E”: "Hfq has been demonstrated to interact with the C-terminal scaffold region of RNase". As mentioned above, it is unclear whether this interaction is direct or occurs via RNA. In fact, one of the papers (Bruce et al.,) cited after this sentence claims that the interaction is dependent on RNA, which is the complete opposite of the authors' statement.

We thank the reviewer for the two points above. It is possible that although the regulation on the selected mRNA does not require the corresponding sRNA regulators, some other RNAs are needed to bridge the Hfq-RNase E interaction. Our data only suggest the Hfq-RNase E interaction is needed, but cannot specify whether RNA is involved in the interaction, which we leave as an open question in the Discussion. We have now clarified the Hfq-RNase E interaction in the Results and Discussion as below:

“The C-terminal region of RNase E serves as a scaffold for the degradosome protein components (RNA helicase RhlB, enolase, and PNPase). Hfq has been demonstrated to interact with the C-terminal scaffold region of RNase E, although it is still under debate whether such interaction is direct or mediated by RNA^28–31^”

“While our results reveal that Hfq binding contributes to mRNA turnover through recruitment of RNase E, more mechanistic details remain to be further elucidated. First, it is under debate whether Hfq-RNase E interaction is direct or mediated by RNAs^28–31^. While our data show that Hfq can promote mRNA degradation without their matching regulatory sRNAs, it remains to be investigated whether other cellular RNAs may participate in bridging the interaction. Second, while our data demonstrate that deletion of the scaffold region of RNase E abolishes the Hfqmediated regulation on mRNA turnover, the same scaffold region also includes the binding sites of other degradomes components, thus we cannot exclude the possibility that these protein components also participate in the regulation. Further experiments are needed to answer these mechanistic questions.”

3) The authors claim that rifampicin "inhibits transcription and results in the loss of most cellular mRNAs". A more correct wording would be that rifampicin treatment inhibits transcription and leads to a global loss of RNA. For instance, many Hfq-binding sRNAs have half-lives shorter than 15 minutes (e.g. Vogel et al., 2003). While the measured increased diffusion of Hfq after rifampicin treatment most likely indicates a higher fraction of Hfq free from RNA, it does not inform on the classes of RNA that contribute to this binding. Either the authors need to explicitly show that mRNAs are preferentially lost over other RNA classes after 15 minutes of rifampicin treatment, or the term "mRNA-associated fraction" used throughout the manuscript should be changed to "RNA-associated fraction".

We thank the reviewer for pointing this out, and indeed for this exact reason, we prefer to term it “mRNA-associated” and “mRNA-free” Hfq, as the diffusion coefficient change is more robustly depending on the presence of mRNA. From the sRNA expression data sets, we can see that the change in diffusion coefficient cannot distinguish sRNA-mRNA-associated Hfq from mRNAassociated Hfq, or sRNA-Hfq from mRNA-free Hfq. However, we think it is fair to name so as sRNA-mRNA-associated form is a subset of the mRNA-associated form, and sRNA-Hfq is a subset of mRNA-free form. Please note that mRNA-free form is not equal to free Hfq. To make this clear in the manuscript, we revised the text as below:

“It should be noted that under the current rifampicin treatment (200 µg/mL concentration for 15 minutes), mRNAs, which have an average half-live of 1-4 min^50^, are preferentially degraded, compared to tRNAs^51^ and rRNAs^52^. While many sRNAs show long half-lives when targetcoupled degradation is reduced in the absence of mRNAs upon rifampicin treatment^53,54^, some sRNAs do have short half-lives^55^. Therefore, rifampicin treatment might also reduce the fraction of Hfq bound by sRNAs. However, our data suggest that binding sRNA to RNA-free Hfq or to mRNA-associated Hfq did not change the diffusion coefficients of corresponding species (see sections below). Therefore, we interpreted that the change in the diffusion coefficient upon rifampicin treatment primarily reflected the binding of mRNAs to Hfq.”

“However, based on the discussion above, it is possible that a subpopulation of mRNA-free or mRNA-associated Hfq might be sRNA-associated Hfq, or sRNA-mRNA-Hfq tertiary complex, respectively.”

4) The authors claim that mutations in the distal phase of Hfq lead to "a large increase in the diffusivity under NT condition". This is a clear overstatement; according to Figure 2C, the increase compared to WT is less than two-fold.

We have revised the sentence as below. “Large” was referring to the comparison with the mutations in distal face or rim.

“..and both distal face mutations (Y25D and K31A) led to the largest increase in the diffusivity under NT condition, with the diffusion coefficients close to the rifampicin treated case …”

5) Subsection “Most Hfq proteins are occupied by mRNAs in the cell during exponential growth”: "Considering the average length of bacterial mRNAs to be 1 kb (~330 kDa), and Mw of bacterial ribosome (~2.5 MDa), this reduction in D supports the interpretation that a significant fraction of WT Hfq proteins are associated with mRNAs in the NT case, and that a fraction of the associated mRNAs are translated by the ribosomes." Do the authors imply that the majority of Hfq is associated with mRNA alone? An alternative explanation would be that most Hfq hexamers are simultaneously bound to mRNA and sRNA. The data does not inform on the nature of the complexes Hfq is involved in, other than that they contain RNA (according to the measured changes in diffusivity during rifampicin treatment). This should be clearly stated in the text.

We thank the reviewer for pointing this out. All three reviewers raised the same issue on the interpretation and discussion on the molecular weight. Our previous interpretation was based on the power-law relationship between molecular weight (M_w_) and the diffusion coefficient (D) (D = aM_w_^x^, where x = -0.33) that was experimentally observed in bacterial cells (reference 42). However, as reviewer 2 suggested, inference of M_w_ just based on D may not be robust, since diffusion coefficient is also dependent on other factors, such as surface charge of the protein. In the revised manuscript, we have removed the part of the discussion, as it does not contribute to our major conclusions of the manuscript.

It is entirely possible that the mRNA-free Hfq contains both free Hfq and sRNA-bound

Hfq, however, the change in the diffusion coefficient caused by sRNA binding compared to free Hfq is not distinguishable. We therefore use “mRNA-free” Hfq to refer to the fraction of Hfq that does not associate with mRNA. Please refer to the response to point (3) above.

6) Subsection “Hfq is deficient in releasing mRNAs without interactions with RNase E”: "The rneΔ14 mutant has a smaller fraction of the C-terminal scaffold (residues 636-845) deleted, encompassing the Hfq binding region." The deleted part of RNase E also encompasses two RNA-binding domains, as well as binding sites for proteins RhlB and enolase. The differences in activity between WT and rneΔ14 could thus stem from impairment of many different interactions and do not specifically report on the loss of a potential interaction with Hfq.

We have revised the text for the description of the *rne*Δ*14* mutant, and added discussion regarding the Hfq-RNase E interaction.

“The *rne*Δ*14* mutant has a smaller fraction of the C-terminal scaffold (residues 636-845) deleted, encompassing the Hfq, RhlB and enolase binding regions and two RNA-binding domains^59^.”

“While our results reveal that Hfq binding contributes to mRNA turnover through recruitment of RNase E, more mechanistic details remain to be further elucidated. First, it is under debate whether Hfq-RNase E interaction is direct or mediated by RNAs^28–31^. While our data show that Hfq can promote mRNA degradation without their matching regulatory sRNAs, it remains to be investigated whether other cellular RNAs may participate in bridging the interaction. Second, while our data demonstrate that deletion of the scaffold region of RNase E abolishes the Hfq mediated regulation on mRNA turnover, the same scaffold region also includes the binding sites of other degradomes components, thus we cannot exclude the possibility that these protein components also participate in the regulation. Further experiments are needed to answer these mechanistic questions.”

7) Subsection “Hfq is deficient in releasing mRNAs without interactions with RNase E”: "In both RNase E mutant backgrounds, the diffusivity of Hfq-mMaple3 became less sensitive to transcription inhibition by rifampicin compared to the WT rne case (Figure 3a and 2c)." The authors should show the data for strains with WT and mutant RNase E in the same graph for easier comparison. In addition, a statistical analysis should be provided to test whether the claimed differences are significant.

We have reanalyzed all the data according to reviewer 2’s suggestion and replotted all the figures for better comparison (new Figure 4 for this specific case). P-values are provided for the comparison.

8) Subsection “Hfq is deficient in releasing mRNAs without interactions with RNase E”: "40-50% of Hfq-mMaple3 remained mRNA associated upon rifampicin treatment in the RNase E mutant backgrounds (Figure 3B)." I could not find the corresponding numbers for the strain with WT RNase E. Please provide these numbers in the figure and/ or in this part of the text for clarity.

We apologize for not being clear in the original manuscript -- this same issue was also brought up by the reviewer 1. In the previous analysis, we assumed that the mRNA-associated fraction is close to zero after rifampicin treatment in the WT rne background. With this assumption, fitting of the rifampicin treated case in the rne131 background generated 40-50% of mRNA-associated fraction. We have now reanalyzed all the data based on reviewer 2’s suggestion. The mRNAbound fractions are reported now for all conditions in the new Figure 2 and Figure 4. The details on the new data analysis method are described in the new Figure 1—figure supplement 3 and Figure 2—figure supplement 2, and summarized in the Essential revisions part 1 above.

9) Subsection “Hfq is deficient in releasing mRNAs without interactions with RNase E”: "These observations suggest that without the Hfq-RNase E interaction, more mRNAs remain bound to Hfq, indicating that Hfq may help deliver the associated mRNA to RNase E for degradation." This is a very far-reaching interpretation of the data presented in Figure 3. There is no evidence for Hfq-mediated delivery of RNase E in Figure 3. I strongly advise the authors to use a more careful interpretation of the data.

We have revised the text as below:

“These observations suggest that without the Hfq-RNase E interaction, more mRNAs remained bound to Hfq, and hint that Hfq-RNase E interaction may help recycle Hfq from the mRNA-associated form through degradation of mRNAs.”

10) In Figure 3C, the authors show half-life measurements of several mRNAs that are regulated by sRNAs. For these experiments, strains with corresponding sRNA gene deletions were used. I assume that the rationale for this was to avoid putative differences due to impaired sRNA regulation. This is not a stringent strategy, as it is unknown whether other sRNAs target these mRNAs. Moreover, from the bar charts in Figure 3C, it appears that mRNA half-lives increase when Hfq carries a distal face mutation. However, in the Northern blots used to create the bar charts (corresponding supplementary figure), there seem to be very small (if any) differences in half-lives between WT and mutant Hfq. For transparency, and easier interpretation for the reader, the authors should (instead of bar charts) plot the log10-transformed relative band intensities versus time, and show all data points (not only error bars). They should also include the corresponding Northern blots in Figure 3 along with the quantifications.

The differences in the mRNA half-life are 50% to 2-fold for the tested three mRNAs, which are indeed not a big difference, but correctly reflect the Northern blot image. We do think there is a very noticeable difference between Y25D and WT Hfq even from the Northern blot. We also include a more stringent strategy as also suggested by reviewer 1, using Hfq Q8A mutant. The detailed description can be found in point 2 of Essential revisions above.

We have moved the Northern blot to the main figure (now in the new Figure 5). In the corresponding figure supplement, we plot individual data points from each replicate, and the mean ± standard deviation calculated from all replicates in log scale. We also show the fitting curves to extract the half-lives. The equation we used for half-life fitting is described in the method as previously published.

11) The authors propose that Hfq, through an interaction with RNase E, promotes mRNA degradation in an sRNA-independent fashion. They also suggest that "mRNA-occupied Hfq proteins are in standby mode for sRNA binding if needed". If this were correct, one would expect that a high cellular concentration of a Class II sRNA, that is a strong competitor for Hfq's interaction with mRNAs, would result not only in displacement of Hfq from many mRNAs, but also thereby increase their stability. ChiX is a strong competitor for distal phase binding that should cause such an effect. The authors interpret their data to imply that ChiX overexpression results in displacement of Hfq from many mRNAs. However, they do not provide evidence that this displacement results in general mRNA stabilization, which should be the outcome according to their model. In fact, induction of ChiX to a high intracellular concentration from a plasmid resulted in downregulation of one specific mRNA (ybfM), rather than a stabilizing effect on many mRNAs (Rasmussen et al., 2009). It is very surprising that the authors did not cite this paper.

We thank the reviewer for the good suggestion. We added new lifetime measurement in the presence of WT ChiX and ChiX with two AAN motif deleted, and found that the presence of WT ChiX can indeed increase the *sdhC* lifetime, similar to Y25D Hfq mutant, whereas the mutant ChiX that has reduced Hfq binding ability was not able to increase the lifetime as significantly as the WT ChiX. The new data is presented as the new Figure 6D. It is worth mentioning that *sdhC* is not a known target of ChiX, therefore the increase in the stability of sdhC is an indirect effect due to competitive binding of ChiX to Hfq. ybfM is a specific target for ChiX, the destabilization of ybfM is a direct result of ChiX-mediated regulation. As in our specific case, since we are not studying ChiX targets, we do not find a good place for citing this particular paper even with the new data.

“As our model suggests that binding of Hfq to the mRNA through the distal face can regulate the mRNA turnover, we reasoned that sRNAs that can effectively compete for Hfq binding against mRNAs may decoy Hfq from this regulatory function. To test this, we again used *sdhC* as an example, and measured its half-life in the presence of ChiX, which is a strong competitor for Hfq binding (Figure 3). In the presence of vector control, *sdhC* exhibited comparable half-life compared to the case without any plasmid (Figure 5D, Figure 6C and D). The presence of WT ChiX increased the half-life by ~70%, whereas the mutant ChiX without two AAN motif deleted only increased the half-life by ~42%, consistent with its reduced binding ability to Hfq (Figure 3A, 3B, Figure 6C and 6D). These results further support our model of Hfq-mediated regulation of mRNA turnover, and demonstrate that the presence of strong Hfq binding sRNAs can modulate the strength of Hfq’s regulation.”

“…we observed that sRNA competitors, such as ChiX, which can outcompete mRNAs for binding at the distal face, can decoy Hfq from regulating mRNA turnover, the same effect as the Y25D mutation. Similar observation was reported previously that ChiX can titrate Hfq from translational repressing transposase mRNA^67^.”

12) Several reports have shown that the 5' moiety of RNA substrates largely influences on both, RNase E cleavage efficiency and specificity of cleavage site selection (e.g. Mackie, 1998, Jiang and Belasco, 2004, Bandyra et al., 2012); RNAs carrying a 5' monophosphate are substantially better substrates than those with a 5' tri-phosphate. Regarding the Hfq-dependent mRNA degradation proposed by the authors, do these mRNAs need to be decapped for RNase E to degrade them? Or does the degradation go through the substantially less efficient internal cleavage route? The authors should discuss both their data and their models with respect to what is known about the cleavage activity/ specificity of RNase E. They should also cite the most seminal papers on this subject.

We thank the reviewer for the good suggestion. We have performed additional experiments to measure the lifetime of *shdC*, as an example. The revised manuscript is quoted as below.

“To test whether this Hfq-mediated mRNA turnover is dependent on the 5’-end decapping, we used *sdhC* as an example, and compared its half-life in the backgrounds of ∆sRNA∆*rppH* and ∆sRNA∆*rppH hfq*Y25D.[…] These results suggest that the decapping by RppH is not required for the Hfq-mediated regulation of mRNA turnover, at least for the case of Hfq regulation on *sdhC* mRNA.”

13) Regarding the EMSA shown in Figure 4. Subsection “sRNAs can displace Hfq from mRNAs in a face-dependent man”: "Results show that RyhB cannot displace the radiolabeled ptsG from Hfq, but rather generates an additional upper-shifted band compared to the band of ptsG-Hfq complex, supporting that RyhB and ptsG can co-occupy Hfq". This is a very creative interpretation of an inconclusive result. What can be deduced from the gel picture is that the band representing the Hfq-ptsG complex becomes weaker at the two highest RyhB concentrations. The reason for this could be either that RyhB displaces ptsG, or that a ternary complex is formed. The design of the experiment makes it impossible to judge whether the former is happening; since the majority of ptsG mRNA is not in complex with Hfq (even in the absence of RyhB), it is impossible to judge whether high concentrations of RyhB results in increased free ptsG mRNA. Regarding the latter possibility (which is put forward by the authors), the "upper-shifted band" is barely visible and do not by any means reach the intensity of the ptsG-Hfq band, which would be the case if the major effect of RyhB addition would be the formation of a ternary complex.

We thank the reviewer for pointing this out. The purpose of the EMSA assay is to show the feasibility of forming ptsG-RyhB-Hfq tertiary complex, as ptsG and RyhB are not matching sRNA-mRNA pair. The purpose of the EMSA is not to quantitatively determine the fraction or efficiency of tertiary complex formation, as it is difficult to fully recapitulate the in vivo condition in an in vitro experiment. But we agree with the reviewer that the quality of the previous gel is not good. We therefore repeated the experiments starting with a higher *ptsG*-Hfq complex to chase with RyhB or ChiX. The new results are presented in the new Figure 3E and 3F. The gel consistently showed the formation of the *ptsG*-RyhB-Hfq complex when chased with RyhB, but not ChiX. However, in the EMSA assay, we also observed direct displacement of *ptsG* fragment by RyhB, which was not indicated by the in vivo imaging results, as our in vivo imaging did not show a shift of Hfq to a fast-diffusing population when overexpressing RyhB. We do not know the exact cause of the discrepancy, but reason that it can largely result from the difference between the cellular conditions and in vitro setting. Nevertheless, the EMSA results still support that class I and class II sRNAs can have different mechanisms to gain access to mRNA-occupied Hfq, and that it is structurally possible to have a class I sRNA co-occupy with a non-target mRNA on Hfq.

14) Subsection “sRNAs can displace Hfq from mRNAs in a face-dependent man”: "In addition, droplet digital PCR (ddPCR) performed in the same conditions as the diffusivity assays showed that RyhB level was comparable to ChiX (Figure 4d)." This is not correct. According to Figure 4d ChiX is almost ten times more abundant than RyhB. The same incorrect statement is repeated later in this subsection.

We thank the reviewer for pointing this out. We added a different normalization method, against the empty vector (results shown in the new Figure 3D). While ChiX level was ~5-fold of the RyhB level when normalized the reads of 16S rRNA , ChiX level was about 50% of RyhB level when normalized to the reads from empty vector (representing the induction fold change). We therefore reason that the difference between ChiX and RyhB when normalizing to the 16S rRNA was very likely due to the different efficiency during RT and PCR steps for these two targets. We have explicitly explained this in the revised manuscript.

15) Subsection “sRNAs can displace Hfq from mRNAs in a face-dependent man”: "EMSA and ddPCR results suggest that both in vitro and in vivo, RyhB can effectively access mRNA-occupied Hfq through co-occupying Hfq from the proximal face." This is an unsubstantiated and probably incorrect interpretation of the results. The EMSA does not provide evidence for a ternary Hfq-ptsG-RyhB complex. In what way does the ddPCR result inform on binding of RyhB to mRNA-bound Hfq?

We apologize for not explaining this well in the previous version. We hope that through the response to point (13), we can convince the reviewer that the EMSA provides in vitro evidence that it is feasible to form Hfq-*ptsG*-RyhB complex. Our logic is as follows: (1) our live-cell tracking results suggest that RyhB expression does not change Hfq diffisity to a fast-diffusing population, in contrast to the case of ChiX, therefore RyhB does not displace mRNA from Hfq effectively. (2) Since RyhB does not displace mRNA from Hfq, it would either be unstable in the cell if it does not bind Hfq at all, therefore showing low cellular abundance, or it can bind to Hfq with mRNA bound at the same time. Using ddPCR and FISH, we showed the RyhB (or SgrS, which shows the same behavior in tracking experiments) level is not significantly lower than ChiX (SgrS level is even higher than ChiX using both normalization methods). We therefore think that RyhB can occupy Hfq together with mRNA. (3) We use EMSA to show the feasibility of forming the tertiary complex, even though the actual fraction of the tertiary complex could be different between in vivo and in vitro experiments. We have significantly revised the manuscript to improve the reasoning here.

“Overexpression of RyhB or SgrS, in contrast, did not cause any significant changes in the HfqmMaple3 diffusivity or the corresponding mRNA-associated fraction (Figure 3A and B). […] Nevertheless, the EMSA results still support that class I and class II sRNAs can have different mechanisms to gain access to mRNA-occupied Hfq, and that it is structurally possible to have a class I sRNA co-occupy with a non-target mRNA on Hfq.”

16) Conceptually, there is no apparent reason why Hfq would not interact with RNA undergoing transcription. A previous Hfq tracking study in *E. coli* indeed reported a three-state model in which the slowest state was interpreted as Hfq bound to RNAs during transcription (Persson et al., 2013). The authors should comment on this finding with regard to their own results. Do the data presented in the current manuscript fit with the previous model? If not, why not?

We have indeed tried to use the same method as in the Nature Methods paper to analyze our data, and got consistent results with their published results on the WT Hfq. Specifically, we also observed a third slowest diffusion state of Hfq, which corresponds to the interpreted Hfq fraction that binds to mRNAs co-transcriptionally. However, due to technical reasons, we decided not to use this analysis method, but used the method as suggested by reviewer 2. For the detailed reason please refer to the Essential revisions part 1. In our method, however, we do not separate Hfq binding to mature mRNAs and mRNA co-transcriptionally. But this does not affect any of our current conclusions.

[Editors’ note: what follows is the authors’ response to the second round of review.]

Essential revisions:However, two critical points still need to be addressed:1) One of the major findings put forward by the authors, which is highlighted both in the abstract and in the schematic Figure 8, is the suggestion that binding of Hfq to mRNAs can recruit RNase E for (sRNA-independent) mRNA degradation. The authors provide live cell Hfq tracking data in strains with mutations in the Hfq distal and proximal phases, combined with full-length or mutant RNase E (Figure 4). The tracking data show that the differences in Hfq diffusion, which are observed between WT and mutant RNase E upon rif treatment, are abolished when Hfq has the distal phase mutation. These data are sound and convincing but do not directly address whether the observed effects are indeed due to mRNA degradation.To directly monitor effects on mRNA degradation, the authors provide Northern blot data monitoring the half-life of three selected mRNAs (Figure 5). According to the authors, and many previous studies, the distal face mutation primarily impairs Hfq-mRNA interactions, while the proximal face mutant primarily impairs Hfq-sRNA interactions, at least when considering Class I sRNAs. In Figure 5, the presented data show that, in the strain expressing WT RNase E, both the distal and the proximal face mutants lead to increased mRNA half-lives. It is well established that sRNA-dependent (Class I sRNAs) regulation requires Hfq to contact both the sRNA and the mRNA. If Hfq-dependent degradation of mRNAs where to be mediated strictly through sRNAs, the increase half-life should be the same for the distal and proximal mutants (binding to both RNAs are required). However, according to the authors, the data presented in Figure 5 shows a greater increase in mRNA half-lives with the distal face mutant compared to the proximal face mutant, and is interpreted as representing Hfq-dependent and sRNA-independent degradation by RNase E. From this follows that the contribution of a sRNA-independent effect on mRNA half-lives can be deduced from the difference between the values obtained in the distal mutant and in the proximal mutant, while the difference between WT Hfq and the distal face mutant is the sum of the sRNA-dependent and sRNA-independent effects on mRNA degradation. In other words, if the authors' hypothesis is correct, there should be a significant difference in mRNA half-life between the distal and proximal face mutants, and this difference should be abolished if Hfq cannot interact with RNase E.Unfortunately, the data provided in Figure 5 do not provide unequivocal evidence for the following reasons: (i) too few replicates: some of the mean values were calculated from only two data points, therefore the standard deviations are not meaningful (ii) there is no information on how many replicates where used for calculating each specific mean value, making it impossible to judge the how reliable specific mean values and standard deviations are, (iii) there is no statistical analysis provided to ensure that the differences are significant (which on the other hand is not possible with only two data points). For these reasons, it is impossible to judge whether the differences between Y25D and Q8A in the WT and rne131 mutant are significant. To test whether the hypothesis is correct, the authors need to provide at least three (four would be advisable) replicates for each mean value, and use statistical tests to assess whether the proposed differences are indeed significant. To increase the possibility for the reader to judge the data, each bar showing mean values in Figure 5D should to be overlaid with the value of each data point used for calculating the mean.

We agree with the reviewers on the interpretation of the data. We have now included 4 replicates for each data set in Figure 5. In addition, we also added 1 or 2 additional replicates for Figure 6 (3 or 4 replicates in total). We have provided p-values for each pairwise comparison. Each bar graph was overlaid with each data point for readers’ judgement. The addition of new replicates changes the mean values slightly, but does not change the conclusions.

2) While the analysis is improved, there appeared to be a misunderstanding on how to use CDF to fit displacement to extract the diffusion coefficients and population percentages. The authors used the CDF fitting of the apparent speed of one step displacement (osd) to extract the average speed and population percentage of Hfq. The authors cited the Yang, 2019 paper as the source. Do note that in Yang, 2019, the authors were measuring the directional moving speed but not random diffusion. For non-processively diffusing molecules, the diffusion coefficient D should be used. While the speed CDF from osd can also be used to extract different populations based on the difference in speed, it is different from classifying molecules based on their apparent diffusion coefficients (note that only osd2, but not osd, is proportional to D). Two molecules diffusing at different speeds (with random orientations) will have different Ds, but the quantitative difference between the two Ds is not the same as that between the two speeds (again, osd2 , but not osd, is proportional to D). Therefore, the classification of fast and slow diffusing molecules based on diffusion coefficient could be different from that based on speed. Furthermore, the mean squared displacement (MSD) measures how far the molecule diffuses away over different time lags. MSD is also an averaged measurement of molecules of different Ds, and not particularly accurate when there are at least two populations of different Ds, which is the case the authors are trying to establish. See https://link.springer.com/protocol/10.1007/978-1-59745-513-8_14 for a review, and Bettridge, et al. https://doi.org/10.1111/mmi.14572, in the supplemental notes for a practical guide. The fractionation of subpopulations based on diffusion coefficient would be important for the authors to make the argument of whether mRNA binding would lower the D or vice. versa. If the difference in speed is fairly large, I suspect that the major conclusions should still hold if the authors switch to the analysis of CDF of osd2, but the correct analysis should be provided.

We thank the reviewer for pointing out the correct way of analyzing the data. As the reviewer wrote, our raw data (one-step displacement) are from random diffusion. Thus, the correct fitting functions for this case are not the error functions (which are for directional motions) as in the previous submission, but the exponential functions as expressed in https://doi.org/10.1111/mmi.14572. All the mRNA-associated fraction plots were updated based on the new fitting, and new fitting values are listed in the Supplementary file 1. Details of the fitting method are presented in the updated Materials and methods section. The new analysis does not change the conclusions of the manuscript.